# Towards Precision Protein-Ligand Affinity Prediction Benchmark: A Complete and Modification-Aware DAVIS Dataset

**Ming-Hsiu Wu[1]**    **Ziqian Xie[1]**    **Shuiwang Ji[2]**    **Degui Zhi[1]**

[1]The University of Texas Health Science Center at Houston
[2]Texas A&M University

{ming.hsiu.wu,ziqian.xie,degui.zhi}@uth.tmc.edu, {sji}@tamu.edu

## Abstract

Advancements in AI for science unlocks capabilities for critical drug discovery tasks such as protein-ligand binding affinity prediction. However, current models overfit to existing oversimplified datasets that does not represent naturally occurring and biologically relevant proteins with modifications. In this work, we curate a complete and modification-aware version of the widely used DAVIS dataset by incorporating 4,032 kinase–ligand pairs involving substitutions, insertions, deletions, and phosphorylation events. This enriched dataset enables benchmarking of predictive models under biologically realistic conditions. Based on this new dataset, we propose three benchmark settings—Augmented Dataset Prediction, Wild-Type to Modification Generalization, and Few-Shot Modification Generalization—designed to assess model robustness in the presence of protein modifications. Through extensive evaluation of both docking-free and docking-based methods, we find that docking-based model generalize better in zero-shot settings. In contrast, docking-free models tend to overfit to wild-type proteins and struggle with unseen modifications but show notable improvement when fine-tuned on a small set of modified examples. We anticipate that the curated dataset and benchmarks offer a valuable foundation for developing models that better generalize to protein modifications, ultimately advancing precision medicine in drug discovery. The benchmark is available at: `https://github.com/ZhiGroup/DAVIS-complete`

## 1 Introduction

Measuring protein-ligand binding affinity is a critical task in drug development, as it directly determines the therapeutic efficacy and selectivity of potential drug candidates [37]. AI breakthrough has revolutionized protein folding [4, 11, 16, 22], protein design [19, 18], and even protein-ligand binding [4, 7, 46, 8]. However, even with breakthroughs like AlphaFold [4, 11, 16] and DiffDock [7, 8], the protein-ligand affinity prediction problem is not solved yet. First, structural predictions from models like AlphaFold are AI estimations, not always equivalent to experimentally solved crystal structures [33, 17]. Second, even with solved protein structures, co-crystalized ligand-bound structures are often unavailable. Third, factors beyond direct structural complementarity, such as pH [30] and solvent effects [3], also significantly influence binding affinity. Current AI-driven affinity prediction methods are still in what might be termed a 'pre-AlphaFold era'; a significant portion operate as 'structure-free' (using 1D protein amino acid sequences) [55, 54, 50, 27] or 'docking-free' even with the incorporation of structural information [25, 15, 44].

A more pressing challenge in the field is the lack of large, diverse, and experimentally homogeneous training datasets [20] that adequately capture biological realities. In particular, protein modifica-

tions—such as substitutions, insertions, deletions, and post-translational modifications (PTMs)—can drastically alter protein structure and ligand interactions [38, 26, 48]. While generating such comprehensive datasets is challenging, current AI-driven models [21, 2, 34, 14, 52, 13, 53, 44, 28] only focus on wild-type proteins, overlooking modified protein versions or applying a simplistic "one-size-fits-all" approach to variants within datasets like DAVIS [10]. This oversight creates a significant gap in understanding how these models perform in real-world biological contexts, where proteins naturally undergo structural or chemical modifications. Models trained solely on such data may overfit to wild-type proteins and fail to generalize to more complex, yet practical, scenarios.

This study aims to bridge this critical gap. We introduce DAVIS-complete, a curated and complete version of the DAVIS dataset [10] that explicitly accounts for protein modifications, as illustrated in Fig. 1(a). Building upon this, we design three novel benchmark frameworks inspired by realistic drug discovery scenarios: (1) Augmented Dataset Prediction: We augment the previously used DAVIS dataset with modified protein–ligand pairs and evaluate model performance across three standard train-test splits, assessing general predictive capability in a diverse setting (Fig. 1(c)). (2) Wild-Type to Modification Generalization: This benchmark assesses a model's ability to generalize from wild-type proteins to unseen modified variants in a zero-shot setting, reflecting practical cases where experimental data for modified proteins are unavailable (Fig. 1(d)). (3) Few-Shot Modification Generalization: We further evaluate model adaptability by fine-tuning on a small number of modified protein–ligand pairs (Fig. 1(e)). This scenario mirrors precision medicine applications, where individualized therapies frequently rely on accurately predicting drug responses from limited genetic or proteomic data unique to each patient. Together, these benchmarks, for the first time, provide a more comprehensive and biologically relevant framework for evaluating binding affinity prediction models.

**Our contributions are:**

- The curation and public release of DAVIS-complete, a comprehensive dataset incorporating protein modifications for binding affinity prediction benchmark.
- The design and proposal of three biologically relevant benchmarks built upon DAVIS-complete.
- An extensive evaluation of existing state-of-the-art methods using these new benchmarks, highlighting current limitations (e.g., overfitting to wild-type proteins) and demonstrating the potential improvement (e.g., through fine-tuning strategies).

## 2 Related Works

### 2.1 Docking free-based models

In scenarios where high-resolution co-crystallized three-dimensional protein structures are unavailable, most existing deep learning approaches for predicting protein-ligand binding affinity operate without considering explicit binding poses—commonly referred to as docking-free methods. These models often represent proteins using amino acid sequences or predicted protein contact maps, while ligands are depicted as SMILES strings or molecular graphs. Deep neural networks are then employed to extract latent features from these representations to predict binding affinities. Notable models in this category include DeepDTA [55], AttentionDTA [54], GraphDTA [27], DGraphDTA [15], and MGraphDTA [50]. These methods have significantly advanced the field by circumventing the high cost of experimentally determining protein–ligand binding conformations. However, due to the nature of their input representations, these models are inherently limited in their ability to capture structural alterations caused by protein mutations or PTMs. This limitation is particularly important, as such modifications frequently occur in biological systems and could substantially influence protein-ligand binding affinity.

### 2.2 Docking-based models

When high-resolution co-crystallized three-dimensional structures are available, docking-based approaches have also made strides in modeling protein-ligand interactions by explicitly considering atom-level interaction details. Unlike docking-free models, these methods incorporate spatial information about the binding pose, allowing for a more accurate depiction of the interaction landscape.

Notable examples include SchNet [36], EGNN [35], and GIGN [51], which leverage 3D convolutional networks or equivariant graph neural networks to process molecular structures and predict binding affinity directly from geometric configurations. However, the applicability of docking-based methods is limited by the availability of high-quality co-crystallized 3D structures.

Building on this direction, the Folding-Docking-Affinity (FDA) [47] provides a framework for binding affinity prediction in scenarios where experimentally determined co-crystallized structures are unavailable. It unifies protein structure prediction, molecular docking, and binding affinity estimation into a single pipeline. FDA employs predicted 3D protein structures (e.g., from AlphaFold [11, 4]) and ligand binding poses generated through docking methods (e.g., DiffDock [7]) to construct realistic protein-ligand complexes at scale. Despite potential noise, FDA explicitly models atom-level interactions within predicted complexes to capture spatial information for binding affinity prediction. In parallel, recent co-folding models that jointly fold proteins and ligands—such as AlphaFold3 [4], Chai-1 [41], Protenix [40], and Boltz-1 [46]—have significantly advanced binding structure generation. Building on this line, Boltz-2 [32] augments its predecessor with an affinity module to predict protein–ligand binding affinity.

## 2.3 Datasets

Two widely used datasets for evaluating the performance of deep learning-based protein-ligand affinity prediction models are DAVIS [10] and KIBA [39]. The DAVIS dataset focuses on kinase–ligand interactions and provides binding affinity values measured as dissociation constants ($K_d$). These measurements offer an experimentally homogeneous, high-quality, and biologically meaningful ground truth, making the dataset suitable for assessing model performance in kinase-targeted drug discovery. On the other hand, the KIBA dataset aggregates various bioactivity measurements, including $K_i$, $K_d$, and IC$_{50}$ values, into a unified KIBA score, providing a broader yet noisier representation of drug-target interactions across a diverse set of kinases and compounds. These datasets have served as the standard benchmarks for docking free-based models like DeepDTA [55], GraphDTA [27], and their variants [54, 50, 25], allowing for consistent performance comparisons across different protein-ligand representation and model architectures.

## 2.4 Related Datasets on Modification-aware Binding

The PSnpBind dataset [5] offers a resource for studying the impact of single-point mutations at protein binding sites on ligand binding affinity. However, its reliance on traditional molecular docking methods may hinder its acceptability, as experimental assays are still the gold standard. The predicted binding conformations are static and may not capture the dynamic, context-dependent effects of mutations. Additionally, the empirical scoring functions used in docking often fail to accurately reflect changes in binding affinity, particularly in mutated proteins [31].

Another large-scale dataset is BindingDB [24], which contains approximately 3 million experimentally measured binding affinity data points, including both modified and unmodified proteins. Despite its scale, the dataset suffers from heterogeneity in assay types, experimental conditions, and reporting formats, leading to inconsistencies that impede data integration and limit its utility for predictive modeling. For example, a study by Landrum et al. [20] have demonstrated that combining IC$_{50}$ or $K_i$ values from different sources introduces significant noise.

In contrast, the DAVIS dataset offers an experimentally homogeneous, high-quality resource on kinase protein–ligand interactions. In addition to wild-type proteins, it also includes numerous data points involving modified kinase proteins. Previous studies using this dataset [55, 54, 27, 15, 50, 25], however, typically treated modified and unmodified kinases as equivalent or excluded the proteins with modifications altogether. Such modifications could significantly affect binding affinity predictions in certain cases. Therefore, indiscriminately incorporating them into predictive models without accounting for their differences-or simply discarding them-may not leveraging the full value of this dataset. Worse, models trained on such oversimplified DAVIS dataset may even overfit the wild-type proteins. Of note, this study aims to curate a complete version of the DAVIS dataset that accounts for all the modified kinases mentioned. Furthermore, the complete dataset is utilized to benchmark previously proposed docking-free methods as well as the recently published docking-based approach, Folding-Docking-Affinity (FDA) [47] and Boltz-2 [32].

# 3 A complete version of DAVIS dataset

The DAVIS dataset [10] covers interactions of 442 kinase proteins with 72 kinase inhibitors. Kinase proteins are represented by their Entrez Gene Symbols and corresponding names, with modification annotations included when applicable. This protein collection primarily focuses on catalytically active human protein kinase domains across the eight major typical kinase groups, representing over 80% of the human protein kinome. The dataset was primarily curated to analyze the selectivity of kinase inhibitors by examining small molecule-kinome interaction patterns. 31,824 binding affinity measurements ($K_d$) were simultaneously determined using a biochemical assay panel developed for this purpose. The consistency of the assay conditions minimizes variations in the experimental settings, which is a common concern found in other heterogeneous datasets [23, 39, 20].

This comprehensive assay provides critical insights into how protein modifications affect kinase binding affinity, which is vital for drug discovery. For example, the T790M mutation in EGFR reduces Lapatinib binding affinity by approximately 360-fold compared to the wild-type, demonstrating the significant impact of single-point mutations. Similarly, kinase conformational states, regulated by phosphorylation, influence inhibitor binding, as seen with Imatinib, a type II inhibitor, which binds more strongly to the inactive conformation of ABL1 kinase than its active state. These findings highlight the importance of considering kinase conformational dynamics in designing targeted therapies.

To include these modified kinase proteins, Entrez Gene Symbols in the dataset were mapped to UniProt IDs, and the corresponding amino acid sequences were retrieved from the UniProt database [1]. We then manually curated 56 modified amino acid sequences for 11 kinase proteins based on available annotations, including substitutions, insertions, deletions, phosphorylations, or any combinations. This process added 4,032 new modified protein-ligand pair data points (56 sequences * 72 ligands) to the dataset. Notable examples include ABL1 variants (e.g., T315I, H396P, F317I) with or without Tyr393 phosphorylation (Fig. 1(b)), EGFR mutations (L858R, T790M), and the FLT3-ITD found in the MV4;11 AML cell line [45]. Moreover, we refined existing entries for 11 kinase proteins (such as JAK, TYK2, and RSK family members) to include annotations of multiple specific domains rather than merely full-length sequences—a distinction also overlooked in previous studies [55, 54, 27, 15, 50], either. We updated these sequences by meticulously selecting domain boundaries based on relevant literature [12] and UniProt annotations [1]. Details of protein modifications are provided in Table. A1.

To formalize our extension of the DAVIS dataset by including modified kinase proteins, we introduce the following notation: Let $P^w = \{p^{w_i} \mid i = 1, 2, 3, \ldots, |P^w|\}$ denote the set of wild-type kinase proteins from the DAVIS dataset. For kinase proteins with modification variants, define: $P^m = \{p^{m_i} \mid i = 1, 2, 3, \ldots, |P^m|\}$ where $P^m$ encompasses all modified variants across proteins, and each $p^{m_i}$ specifically denotes the set of modified variants for a given protein. Each modified kinase variant within $p^{m_i}$ is represented by $p_j^{m_i}$, corresponding to a specific type of modification (e.g., mutation, deletion, post-translational modification, or combinations thereof). Thus: $p^{m_i} = \{p_j^{m_i} \mid j = 1, 2, 3, \ldots |p^{m_i}|\}$, $p^{m_i}$ can be the empty set if no modified variants are available in the dataset. To denote the combined set including both wild-type and modified kinase proteins, we introduce: $P^* = P^w \cup P^m$. The ligand set is denoted as: $L = \{l_k \mid k = 1, 2, 3, \ldots, |L|\}$, where each ligand $l_k$ represents a distinct chemical compound in the dataset. $A(p, l)$ denotes the binding affinity between a protein $p$ and a ligand $l$.

The DAVIS affinity distribution is dominated by capped measurements: approximately 70% of pairs are reported at $K_d > 10\mu M$ ($pK_d = 5$), over-representing weaker interactions. Among uncapped values, affinities center at $pK_d = 6.48 \pm 1.05$; median 6.24; IQR 5.68–7.08; range $[5.00, 10.80]$). To assess modification effects, we quantify the affinity alternation $\Delta pK_d = A(p_j^{m_i}, l_k) - A(p^{w_i}, l_k)$, which has mean $-0.21 \pm 0.84$; median -0.04; IQR -0.59–0.30; range $[-4.49, 3.02]$. However, due to the $10\mu M$ cap, the exact $\Delta pK_d$ is unobservable for 60% of modified pairs, where the WT, the modification, or both exceed this threshold. Additional details are provided in Appendices B and C.

# 4 Benchmark Design

We aim to assess whether the proposed state-of-the-art deep learning models can accurately predict binding affinity by distinguishing subtle differences among protein modifications. To reflect realistic

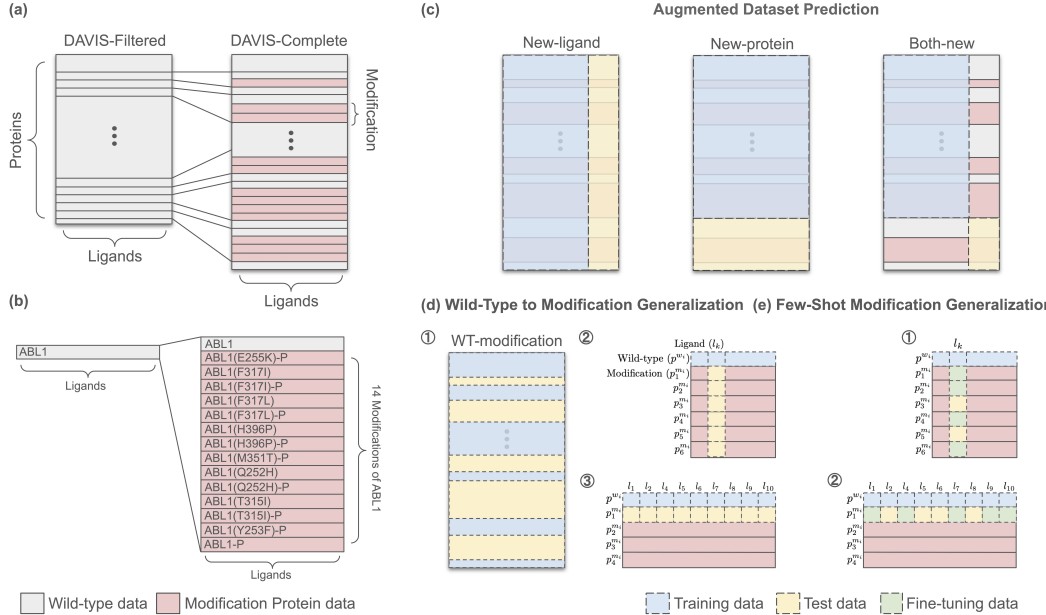

Figure 1: (a) DAVIS-Complete is curated by adding modified kinase protein–ligand pairs previously excluded from DAVIS-Filtered. (b) Example of dataset extension: 14 modifications of the kinase ABL1 are incorporated alongside its wild-type form. (c) Augmented Dataset Prediction benchmark: Wild-type and modified protein–ligand pairs are combined and evaluated under three main splits—new-drug, new-protein, and both-new—each with corresponding sub-splits. (d) Wild-Type to Modification Generalization benchmark: models trained on wild-type pairs are evaluated across (1) global modification generalization, (2) same-ligand different-modifications, and (3) same-modification different-ligands. (e) Few-Shot Modification Generalization: models fine-tuned on limited modified pairs to assess generalization to unseen variants.

drug discovery scenarios, we design three distinct benchmarking settings—summarized in Table 1—to evaluate the model's predictive performance.

**Augmented Dataset Prediction**  We augment the DAVIS dataset used in prior studies [55, 54, 27, 15, 50, 25] by adding modified proteins that were previously ignored. Following prior work [25, 47], we evaluate model performance under three main train–test splits (Figure 1(c)). In all cases, both wild-type and modified protein–ligand pairs are included and mixed in the training and test sets, denoted as $P^*L$. Each main split has corresponding sub-splits. For the new-ligand split, the ligand-name setting ensures no ligand name overlaps between training and test sets, whereas the stricter ligand-structure setting requires that ligands in the test set have a Tanimoto similarity $\leq 0.5$ (computed using Morgan fingerprints) to any ligand in the training set. For the new-protein split, the protein-modification setting treats different modification variants of the same kinase as distinct unseen proteins (e.g., training on ABL1(Q252H) and testing on ABL1(T315I)); the protein-name setting excludes all variants (including wild-type) of a protein from the test set if any variant appears in training; and the protein-seqid setting, the strictest version, ensures that kinases in the training set share $\leq 50\%$ sequence identity with any kinase in the test set. Combining the new-ligand and new-protein strategies yields six both-new configurations. Our benchmark includes the most lenient (ligand-name & protein-modification) and the strictest (ligand-structure & protein-seqid) configurations. The train/validation/test split ratio is kept as close as possible to 70%/10%/20%. The details of model training can be found in Table. A3. Binding affinity prediction performances are evaluated using mean squared error (MSE) and Pearson correlation coefficient ($R_p$).

**Wild-Type to Modification Generalization**  To assess how well models transfer binding-affinity prediction from wild type to modified proteins, we train each model exclusively on wild-type protein–ligand pairs ($P^wL$) and evaluate under three biologically motivated settings. We report MSE, $R_p$, and C-index, and compare against two informative baselines: (i) wild-type ground truth

($y_{\text{WT}}$), which predicts a modified pair's affinity by reusing the measured affinity of its corresponding wild-type pair; and (ii) wild-type prediction ($\hat{y}_{\text{WT}}$), which reuses the model's prediction for the wild-type pair. Models that do not surpass these baselines fail to capture modification-specific effects beyond what is already implied by the wild type. The evaluation settings are: (1) Global modification generalization: The model is evaluated on a broad set of modified protein-ligand pairs (Figure 1(d-1)). It reflects the challenge of predicting binding affinity across diverse protein variants arising from genetic mutations, deletions, or PTM—common in cancer, infectious diseases, and personalized medicine contexts. (2) Same-ligand, different-modifications: The model is tested on multiple distinct modifications of a single kinase, all bound to the same ligand (Figure 1(d-2)). This setting mimics drug resistance studies, where a therapeutic compound must be evaluated across different mutation profiles of a known target protein (e.g., EGFR inhibitors in lung cancer [9, 29]). (3) Same-modification, different ligands: The model predicts binding affinity for a set of ligands against a single modified kinase (Figure 1(d-3)). This scenario supports modification-specific drug screening, where the goal is to identify new compounds that effectively bind a disease-relevant mutant protein and potentially overcome resistance to existing therapies. Appendix G provides further details for the baseline calculation.

**Few-Shot Modification Generalization** Building on the same-ligand, different-modifications and same-modification, different-ligands scenarios from the Wild-Type to Modification Generalization benchmark, we further examine model adaptability by fine-tuning on a limited set of modified protein-ligand pairs, as illustrated in Figure. 1(e). 80% of the available modified protein–ligand pairs are used for model fine-tuning, and the remaining 20% for evaluation. The details of model fine-tuning can be found in Table. A4. This few-shot generalization scenario closely aligns with precision medicine contexts, where personalized treatments often depend on accurately predicting drug responses from sparse, patient-specific genetic or proteomic data. Enhancing model performance in such settings is essential for effectively guiding individualized therapeutic decisions.

Table 1: Summary of benchmarks, sub-tasks, and dataset splits. Training, fine-tuning, and test sets are represented using the introduced notations.

| Benchmark | Sub-task | Training | Fine-tuning | Test |
|---|---|---|---|---|
| Augmented Dataset Prediction | New-ligand | $P^*L$ | - | $P^*L'$ |
| | New-protein | $P^*L$ | - | $P^{*'}L$ |
| | Both-new | $P^*L$ | - | $P^{*'}L'$ |
| Wild-Type to Modification Generalization | Global modification generalization | $P^wL$ | - | $P^mL$ |
| | Same-ligand, different-modifications | $P^wL$ | - | $p^{m_i}l_k$ |
| | Same-modification, different-ligands | $P^wL$ | - | $p_j^{m_i}L$ |
| Few-Shot Modification Generalization | Same-ligand, different-modifications | $P^wL$ | $p_j^{m_i}l_k$ | $p_{j'}^{m_i}l_k$ |
| | Same-modification, different-ligands | $P^wL$ | $p_j^{m_i}l_k$ | $p_j^{m_i}l_{k'}$ |

# 5 Experiments

We benchmark five docking-free models—DeepDTA [55], AttentionDTA [54], GraphDTA [27], DGraphDTA [15], and MGraphDTA [50]—and two docking-based models, FDA [47] and Boltz-2 [32], on the curated, complete DAVIS dataset. Details of input preprocessing and all models are provided in Appendices D-F. Our evaluation covers following three benchmarks:

## 5.1 Augmented Dataset Prediction

We define seven train–test split settings to evaluate prediction performance: ligand-name, ligand-structure, protein-modification, protein-name, protein-seqid, ligand-name & protein-modification, and ligand-structure & protein-seqid. Table 2 reports results on the complete test set, as well as on two subsets: one containing wild-type protein–ligand pairs (wild-type subset) and the other containing modified protein–ligand pairs (modification subset). Across all splits except protein-modification and protein-name, the docking-based methods consistently outperforms docking-free models, achieving higher $R_p$ and lower MSE values. Within the docking-based group, Boltz-2 significantly outperforms FDA, with mean improvements of about 0.11 lower MSE and 0.13 higher $R_p$ overall. This trend holds for both the wild-type and modification subsets. The biggest gains occur on ligand-structure

& protein-seqid, especially in the Modification subset ($\Delta$ MSE -0.42, $\Delta R_p$ +0.29). In the protein-modification split, all models generally demonstrate stronger performance compared to other train-test splits (MSE < 0.5, $R_p$ > 0.6). However, on the complete test set and the modification subset, Boltz-2 no longer holds the top rank; it is surpassed by the simplest docking-free baseline, DeepDTA.

The observation from these splits suggests that binding affinity prediction performance depends strongly on whether proteins or ligands are seen during training. For new-ligand tests, $R_p$ is consistently higher under the ligand-name split than under the stricter ligand-structure split, reflecting the added difficulty of enforcing structural novelty. In the new-protein splits, similarly, performance declines as the test proteins become more dissimilar to those in training (protein-modification $\rightarrow$ protein-name $\rightarrow$ protein-seqid). Comparing new-ligand and new-protein splits, models generally perform worse in new-ligand, indicating a higher dependency on ligand familiarity. Overall, models perform worse in the both-new setting than in the corresponding new-ligand or new-protein splits, with the strictest ligand-structure & protein-seqid configuration yielding the lowest performance.

In particular, we found that this dependency is even more evident among docking-free methods. By examining the degree of performance decline across different splits, it is clear that docking-free models suffer sharper drops in accuracy when proteins, ligands, or both are not present in the training data, highlighting their stronger reliance on seen training examples, which is consistent with previous findings [43, 6, 49, 42]. Besides, when proteins and ligands are included in the training set, most of docking-free methods consistently outperform the docking-based model. This suggests that docking-free approaches may be better at learning direct mappings between known protein-ligand pairs and their binding affinities, whereas the docking-based model, which relies on binding conformation, may not benefit as much from the simple presence of proteins or ligands in the training phase.

## 5.2   Wild-Type to Modification Generalization

To assess the models' ability to generalize from wild-type to modified kinase proteins, we train each model exclusively on wild-type protein-ligand pairs ($P^w L$). We then evaluate their performance across three distinct test scenarios. In the first scenario, termed Global modification generalization ($P^m L$), all modified kinase proteins are included. We additionally stratify results into four subsets defined by whether the wild-type (WT) and modification affinities are capped or uncapped (Details in Appendix C). Results are reported in Table 3. In the WT-uncapped & modification-uncapped subset, DeepDTA, AttentionDTA, DGraphDTA, and MGraphDTA perform similarly well on MSE, $R_p$, and C-index, while GraphDTA, FDA, and Boltz-2 lag behind. However, for these docking-free models the predictions for modification pairs are highly correlated with their own WT predictions (high $R_p(\hat{y}, \hat{y}_{\text{WT}})$), whereas this correlation is much lower for the docking-based FDA. Notably, about 84% of affinity changes lie within $[-1, 1]$ in this category (Fig. A3(a)). For the stronger docking-free models, $R_p$ is nearly identical to $R_p(y, \hat{y}_{\text{WT}})$, indicating overfitting to WT: because the modification–WT differences are majorly small, simply echoing the seen WT prediction yields seemingly strong performance. By contrast, when WT is capped (WT-capped & modification-uncapped) or the modification is capped (WT-uncapped & modification-capped), models can no longer rely on guessing the WT value. For the WT-overfitting models, both $R_p$ and C-index drop in the former; MSE rises in the latter. The docking based, FDA and Boltz-2, becomes relatively stronger. Finally, in the WT-capped & modification-capped subset, WT overfitting docking-free models again appear to perform well, mirroring the pattern observed in the WT-uncapped & modification-uncapped case.

Furthermore, in real-world biological scenarios, a kinase protein often exhibits multiple distinct mutations across different populations, potentially leading to varied binding affinities for the same ligand. To capture this biologically relevant variability, we introduce a second evaluation scenario—same-ligand, different-modifications—to examine whether models pre-trained solely on wild-type proteins can effectively distinguish variations in binding affinity caused by diverse protein modifications when interacting with the same ligand. Notably, in contrast to the global setting that mixes multiple ligands and kinases, this benchmark isolates a fixed kinase–ligand pair ($p^{m_i} l_k$) and restricts evaluation to WT-uncapped & modification-uncapped pairs, varying only the kinase modification to test fine-grained sensitivity. Results are shown in Table 4(a). In terms of MSE, DeepDTA, AttentionDTA, DGraphDTA, and MGraphDTA perform comparably, whereas GraphDTA, FDA, and Boltz-2 show weaker performance. Notably, only MGraphDTA nominally exceeds the $y_{\text{WT}}$ baseline; however, given the large standard deviation, this difference is not meaningful. Simply using the wild-type

ground-truth affinity ($y_{\text{WT}}$) matches or exceeds these models. Furthermore, the consistently low $R_p$ (below 0.3) and marginally better-than-random C-index (just above 0.5) suggest that current models fail to capture or generalize protein modifications from the wild-type training data. Among these approaches, docking-free methods fare worse than docking-based ones. A case study on EGFR variants with staurosporine (Fig. A4) illustrates this: the docking-free MGraphDTA overfits to the wild type and produces nearly identical predictions across variants, whereas the docking-based FDA better tracks the affinity trends.

In another biologically relevant scenario, we may need to rank different ligands for a modified protein. To assess this, we introduce the third scenario—same-modification, different-ligands—which tests whether models trained only on wild-type proteins can distinguish ligand affinities for the same modified kinase. This benchmark fixes a kinase–ligand pair ($p^{m_i}l_k$) and also evaluates only WT-uncapped & modification-uncapped cases, varying only the ligand to probe sensitivity. The results of this evaluation are summarized in Table 4(b). The models perform notably better in the same-modification, different-ligands scenario compared to the same-ligand, different-modifications setting, particularly in terms of $R_p$ and C-index. However, in the case of EGFR(L858R, T790M) with various ligands (Fig. A5), MGraphDTA predictions closely follow the binding affinity trend of the wild-type, again reflecting its tendency to overfit to wild-type data. Additionally, in most cases (44 out of 55), such as EGFR(G719C) (Fig. A6), we observe strong consistency between wild-type and modified protein-ligand affinity profiles, with $R_p$ values above 0.8. This suggests that ligand often plays a more dominant role than protein modification, and the effect of modification on binding affinity is generally smaller. Consequently, the WT–overfitting docking-free models can still outperform the docking-based method in this scenario. Nonetheless, a model's ability to surpass the $y_{\text{WT}}$ baseline remains a meaningful indicator of its sensitivity to subtle affinity shifts.

Table 2: Performance comparison of docking-free and docking-based methods on the complete test set, wild-type subset, and modification subset across seven train–test splits. Results are reported as mean (std) over five random splits. Pearson correlation coefficient ($R_p$) and Mean Squared Error (MSE) are computed from predicted vs. true $pK_d$ values.

| Model | New-ligand | | | | New-protein | | | | | | Both-new | | | |
|---|---|---|---|---|---|---|---|---|---|---|---|---|---|---|
| | Ligand-name | | Ligand-structure | | Protein-modification | | Protein-name | | Protein-seqid | | Ligand-name & Protein-modification | | Ligand-structure & Protein-seqid | |
| | MSE ↓ | $R_p$ ↑ | MSE ↓ | $R_p$ ↑ | MSE ↓ | $R_p$ ↑ | MSE ↓ | $R_p$ ↑ | MSE ↓ | $R_p$ ↑ | MSE ↓ | $R_p$ ↑ | MSE ↓ | $R_p$ ↑ |
| **Complete Test Set** | | | | | | | | | | | | | | |
| DeepDTA | 0.71 (0.11) | 0.31 (0.05) | 0.69 (0.08) | 0.26 (0.07) | **0.29** (0.03) | **0.81** (0.02) | 0.38 (0.06) | 0.74 (0.04) | 0.54 (0.12) | 0.68 (0.02) | 0.77 (0.12) | 0.30 (0.04) | 0.97 (0.14) | 0.12 (0.10) |
| AttentionDTA | 0.71 (0.09) | 0.29 (0.09) | 0.71 (0.10) | 0.26 (0.07) | 0.32 (0.03) | 0.79 (0.02) | 0.37 (0.04) | 0.74 (0.02) | 0.59 (0.15) | 0.64 (0.04) | 1.00 (0.18) | 0.27 (0.10) | 0.89 (0.13) | 0.26 (0.10) |
| GraphDTA | 0.79 (0.14) | 0.30 (0.11) | 0.85 (0.15) | 0.15 (0.11) | 0.39 (0.05) | 0.74 (0.02) | 0.45 (0.06) | 0.67 (0.06) | 0.71 (0.13) | 0.53 (0.06) | 0.87 (0.15) | 0.24 (0.09) | 1.07 (0.27) | 0.08 (0.15) |
| DGraphDTA | 0.71 (0.16) | 0.22 (0.14) | 0.76 (0.08) | 0.10 (0.10) | 0.41 (0.05) | 0.73 (0.02) | 0.46 (0.06) | 0.67 (0.03) | 0.73 (0.11) | 0.50 (0.06) | 0.85 (0.13) | 0.23 (0.05) | 0.98 (0.17) | -0.05 (0.04) |
| MGraphDTA | 0.68 (0.09) | 0.34 (0.08) | 0.80 (0.18) | 0.28 (0.08) | 0.32 (0.04) | 0.79 (0.02) | 0.39 (0.05) | 0.72 (0.04) | 0.63 (0.10) | 0.60 (0.06) | 0.81 (0.13) | 0.33 (0.09) | 0.97 (0.16) | 0.15 (0.08) |
| FDA | 0.60 (0.13) | 0.42 (0.07) | 0.66 (0.08) | 0.36 (0.10) | 0.33 (0.02) | 0.78 (0.01) | **0.36** (0.04) | **0.75** (0.02) | 0.49 (0.09) | 0.70 (0.01) | 0.59 (0.15) | 0.48 (0.04) | 0.89 (0.13) | 0.28 (0.07) |
| Boltz-2 | **0.47** (0.09) | **0.61** (0.06) | **0.50** (0.06) | **0.57** (0.05) | 0.31 (0.04) | 0.80 (0.03) | **0.36** (0.04) | **0.75** (0.03) | **0.47** (0.05) | **0.74** (0.03) | **0.45** (0.13) | **0.63** (0.06) | **0.62** (0.07) | **0.58** (0.07) |
| **Wild-type Subset** | | | | | | | | | | | | | | |
| DeepDTA | 0.60 (0.09) | 0.26 (0.06) | 0.60 (0.07) | 0.23 (0.08) | **0.30** (0.03) | **0.75** (0.01) | **0.31** (0.03) | 0.74 (0.03) | 0.44 (0.06) | 0.67 (0.03) | 0.69 (0.14) | 0.23 (0.06) | 0.78 (0.13) | 0.10 (0.08) |
| AttentionDTA | 0.60 (0.08) | 0.24 (0.08) | 0.62 (0.09) | 0.23 (0.05) | 0.33 (0.03) | 0.72 (0.01) | 0.32 (0.02) | 0.73 (0.01) | 0.47 (0.09) | 0.64 (0.04) | 0.92 (0.16) | 0.20 (0.11) | 0.75 (0.14) | 0.17 (0.08) |
| GraphDTA | 0.66 (0.13) | 0.27 (0.11) | 0.73 (0.14) | 0.11 (0.10) | 0.38 (0.04) | 0.68 (0.01) | 0.38 (0.03) | 0.66 (0.03) | 0.54 (0.05) | 0.56 (0.02) | 0.74 (0.16) | 0.19 (0.06) | 0.90 (0.29) | 0.03 (0.13) |
| DGraphDTA | 0.58 (0.14) | 0.20 (0.14) | 0.66 (0.07) | 0.05 (0.09) | 0.43 (0.05) | 0.63 (0.02) | 0.42 (0.04) | 0.63 (0.02) | 0.61 (0.05) | 0.49 (0.02) | 0.72 (0.14) | 0.14 (0.08) | 0.78 (0.14) | -0.05 (0.04) |
| MGraphDTA | 0.58 (0.07) | 0.30 (0.10) | 0.69 (0.15) | 0.23 (0.06) | 0.34 (0.04) | 0.72 (0.02) | 0.34 (0.03) | 0.71 (0.02) | 0.51 (0.06) | 0.60 (0.02) | 0.68 (0.17) | 0.26 (0.10) | 0.79 (0.14) | 0.12 (0.05) |
| FDA | 0.53 (0.12) | 0.35 (0.10) | 0.59 (0.08) | 0.30 (0.09) | 0.32 (0.03) | 0.72 (0.01) | **0.31** (0.02) | **0.74** (0.01) | 0.41 (0.04) | 0.69 (0.01) | 0.53 (0.17) | 0.38 (0.03) | 0.76 (0.13) | 0.21 (0.07) |
| Boltz-2 | **0.42** (0.08) | **0.55** (0.07) | **0.46** (0.07) | **0.52** (0.05) | **0.30** (0.04) | **0.75** (0.03) | 0.32 (0.03) | 0.73 (0.02) | **0.42** (0.04) | **0.72** (0.02) | **0.41** (0.15) | **0.56** (0.09) | **0.54** (0.06) | **0.53** (0.07) |
| **Modification Subset** | | | | | | | | | | | | | | |
| DeepDTA | 1.52 (0.31) | 0.30 (0.09) | 1.34 (0.20) | 0.25 (0.10) | **0.21** (0.06) | **0.94** (0.02) | 0.79 (0.35) | 0.70 (0.13) | 0.88 (0.37) | 0.66 (0.04) | 1.35 (0.18) | 0.37 (0.09) | 1.67 (0.55) | -0.02 (0.20) |
| AttentionDTA | 1.49 (0.29) | 0.31 (0.17) | 1.36 (0.27) | 0.29 (0.14) | 0.22 (0.08) | 0.93 (0.02) | 0.71 (0.13) | 0.74 (0.07) | 0.99 (0.33) | 0.56 (0.15) | 1.48 (0.47) | 0.38 (0.20) | 1.43 (0.41) | 0.30 (0.15) |
| GraphDTA | 1.67 (0.27) | 0.25 (0.14) | 1.66 (0.22) | 0.12 (0.14) | 0.47 (0.18) | 0.86 (0.04) | 0.87 (0.34) | 0.65 (0.15) | 1.15 (0.31) | 0.43 (0.17) | 1.74 (0.30) | 0.24 (0.12) | 1.74 (0.61) | 0.03 (0.28) |
| DGraphDTA | 1.63 (0.38) | 0.18 (0.18) | 1.47 (0.25) | 0.12 (0.11) | 0.25 (0.13) | 0.93 (0.03) | 0.81 (0.28) | 0.73 (0.10) | 1.21 (0.41) | 0.45 (0.23) | 1.76 (0.34) | 0.27 (0.06) | 1.64 (0.47) | -0.12 (0.08) |
| MGraphDTA | 1.43 (0.26) | 0.36 (0.09) | 1.56 (0.53) | 0.29 (0.18) | 0.22 (0.07) | 0.93 (0.02) | 0.73 (0.23) | 0.72 (0.10) | 1.12 (0.29) | 0.52 (0.25) | 1.64 (0.46) | 0.38 (0.16) | 1.61 (0.54) | 0.13 (0.16) |
| FDA | 1.11 (0.23) | 0.53 (0.07) | 1.15 (0.23) | 0.45 (0.13) | 0.39 (0.08) | 0.88 (0.02) | 0.71 (0.16) | 0.74 (0.07) | 0.74 (0.26) | 0.71 (0.03) | 0.95 (0.16) | 0.60 (0.06) | 1.37 (0.29) | 0.32 (0.10) |
| Boltz-2 | **0.87** (0.14) | **0.70** (0.05) | **0.77** (0.13) | **0.67** (0.07) | 0.36 (0.05) | 0.89 (0.02) | **0.63** (0.15) | **0.76** (0.08) | **0.65** (0.22) | **0.74** (0.07) | **0.75** (0.15) | **0.73** (0.05) | **0.95** (0.25) | **0.61** (0.09) |

## 5.3 Few-Shot Modification Generalization

In the same-ligand, different-modifications setting, the evaluation results are summarized in Table. 5(a), which reports model performance before ($\hat{y}$) and after fine-tuning ($\hat{y}_{\text{FT}}$) across three metrics: MSE, $R_p$, and C-index. All docking-free models show improved performance after fine-tuning, demonstrating the value of even limited modified kinase data in enhancing generalization at the protein modification level. However, in terms of $R_p$ and C-index, all models still exhibit low performance—remaining below or equal 0.6, which is often considered the threshold for effective prediction.

In the same-modification, different-ligands setting, the benchmark results are shown in Table. 5(b). All models except the docking-based FDA model similarly show noticeable improvements across evaluation metrics—MSE, $R_p$, and C-index—after fine-tuning on few-shot samples. Among all models, AttentionDTA achieves the best overall performance, with its MSE decreasing from 0.62 to 0.42, $R_p$ increasing from 0.77 to 0.80, and C-index improving from 0.81 to 0.82. These results

Table 3: Performance comparison of docking-free models and a docking-based method trained exclusively on wild-type protein–ligand pairs ($P^w L$) and evaluated on modified kinase protein–ligand pairs ($P^m L$). The $P^m L$ test set is partitioned into four distinct subsets depending on whether affinity values are capped or not. Results are reported as mean (standard deviation) over five independent runs using identical train–test splits but different model parameter initialization. MSE, $R_p$, and C-index are computed between predicted and true $pK_d$ values.

| Model | MSE ↓ | $R_p$ ↑ | C-index ↑ | $R_p(y, \hat{y}_{WT})$ | $R_p(\hat{y}, \hat{y}_{WT})$ | MSE ↓ | $R_p$ ↑ | C-index ↑ | $R_p(y, \hat{y}_{WT})$ | $R_p(\hat{y}, \hat{y}_{WT})$ |
|---|---|---|---|---|---|---|---|---|---|---|
| | **WT-uncapped & modification-uncapped** | | | | | **WT-capped & modification-uncapped** | | | | |
| DeepDTA | 0.63 (0.04) | 0.79 (0.01) | 0.79 (0.01) | 0.79 (0.01) | 1.00 (0.00) | 0.35 (0.01) | 0.11 (0.05) | 0.53 (0.03) | 0.08 (0.02) | 0.87 (0.20) |
| AttentionDTA | 0.66 (0.07) | **0.80** (0.02) | **0.80** (0.01) | 0.79 (0.02) | 0.99 (0.01) | 0.37 (0.01) | 0.06 (0.04) | 0.50 (0.02) | 0.05 (0.04) | 0.84 (0.20) |
| GraphDTA | 1.17 (0.12) | 0.64 (0.02) | 0.74 (0.01) | 0.76 (0.03) | 0.87 (0.02) | 0.34 (0.03) | 0.00 (0.07) | 0.51 (0.02) | 0.03 (0.07) | 0.89 (0.09) |
| DGraphDTA | 0.64 (0.02) | **0.80** (0.01) | **0.80** (0.00) | 0.80 (0.00) | 0.97 (0.04) | 0.33 (0.01) | 0.03 (0.06) | 0.51 (0.02) | 0.04 (0.06) | 0.97 (0.04) |
| MGraphDTA | **0.61** (0.04) | **0.80** (0.01) | **0.80** (0.01) | 0.79 (0.01) | 0.99 (0.01) | 0.37 (0.01) | 0.05 (0.07) | 0.54 (0.03) | 0.02 (0.08) | 0.92 (0.09) |
| FDA | 1.47 (0.05) | 0.62 (0.01) | 0.72 (0.00) | 0.78 (0.02) | 0.58 (0.02) | 0.30 (0.01) | 0.13 (0.02) | 0.53 (0.01) | 0.07 (0.07) | 0.09 (0.09) |
| Boltz-2 | 0.83 (0.12) | 0.75 (0.04) | 0.77 (0.02) | 0.73 (0.05) | 0.92 (0.02) | **0.24** (0.04) | **0.20** (0.05) | **0.58** (0.01) | 0.13 (0.06) | 0.70 (0.14) |
| | **WT-uncapped & modification-capped** | | | | | **WT-capped & modification-capped** | | | | |
| DeepDTA | 1.89 (0.20) | – | – | – | 0.99 (0.00) | **0.01** (0.00) | – | – | – | 0.96 (0.03) |
| AttentionDTA | 2.06 (0.40) | – | – | – | 0.93 (0.13) | **0.01** (0.01) | – | – | – | 0.92 (0.04) |
| GraphDTA | 1.76 (0.14) | – | – | – | 0.99 (0.00) | 0.03 (0.01) | – | – | – | 0.67 (0.03) |
| DGraphDTA | 1.92 (0.14) | – | – | – | 1.00 (0.00) | **0.01** (0.00) | – | – | – | 0.98 (0.00) |
| MGraphDTA | 1.51 (0.14) | – | – | – | 0.90 (0.04) | **0.01** (0.00) | – | – | – | 0.95 (0.04) |
| FDA | 0.65 (0.03) | – | – | – | 0.50 (0.05) | 0.05 (0.00) | – | – | – | 0.11 (0.03) |
| Boltz-2 | **0.57** (0.06) | – | – | – | 0.60 (0.06) | 0.05 (0.03) | – | – | – | 0.80 (0.07) |

Table 4: Wild-Type to Modification Generalization benchmark (a) Same-ligand, different-modifications: models are trained on all wild-type pairs and evaluated on modified variants of the same kinase protein with a fixed ligand. (b) Same-modification, different-ligands: models are evaluated on distinct ligands for a fixed kinase modification. Metrics are mean (std) across kinase–ligand combinations.

| Model | MSE ↓ | | | $R_p$ ↑ | | | C-index ↑ | | |
|---|---|---|---|---|---|---|---|---|---|
| | $y_{WT}$ | $\hat{y}_{WT}$ | $\hat{y}$ | $y_{WT}$ | $\hat{y}_{WT}$ | $\hat{y}$ | $y_{WT}$ | $\hat{y}_{WT}$ | $\hat{y}$ |
| | **(a) Same-ligand, different-modifications** | | | | | | | | |
| DeepDTA | 0.61 (0.72) | 0.63 (0.59) | 0.62 (0.57) | – | – | 0.10 (0.31) | – | – | 0.53 (0.11) |
| AttentionDTA | 0.61 (0.72) | 0.68 (0.76) | 0.65 (0.73) | – | – | 0.10 (0.28) | – | – | 0.53 (0.10) |
| GraphDTA | 0.61 (0.72) | 0.73 (0.68) | 1.07 (1.46) | – | – | -0.02 (0.31) | – | – | 0.50 (0.12) |
| DGraphDTA | 0.61 (0.72) | **0.61** (0.65) | 0.62 (0.64) | – | – | 0.03 (0.26) | – | – | 0.52 (0.11) |
| MGraphDTA | 0.61 (0.72) | **0.61** (0.63) | **0.59** (0.61) | – | – | 0.11 (0.32) | – | – | 0.53 (0.12) |
| FDA | 0.61 (0.72) | 0.83 (1.04) | 1.41 (1.89) | – | – | 0.18 (0.46) | – | – | 0.56 (0.18) |
| Boltz-2 | 0.61 (0.72) | 0.81 (0.75) | 0.79 (0.81) | – | – | **0.29** (0.41) | – | – | **0.60** (0.16) |
| | **(b) Same-modification, different-ligands** | | | | | | | | |
| DeepDTA | 0.53 (0.59) | 0.58 (0.49) | 0.56 (0.47) | 0.86 (0.16) | **0.84** (0.16) | 0.84 (0.15) | 0.84 (0.08) | **0.83** (0.08) | 0.83 (0.07) |
| AttentionDTA | 0.53 (0.59) | 0.62 (0.57) | 0.59 (0.53) | 0.86 (0.16) | **0.84** (0.16) | 0.84 (0.15) | 0.84 (0.08) | **0.83** (0.08) | **0.84** (0.08) |
| GraphDTA | 0.53 (0.59) | 0.90 (0.66) | 1.26 (1.08) | 0.86 (0.16) | 0.79 (0.15) | 0.70 (0.26) | 0.84 (0.08) | 0.82 (0.06) | 0.79 (0.09) |
| DGraphDTA | 0.53 (0.59) | 0.58 (0.50) | 0.58 (0.49) | 0.86 (0.16) | **0.84** (0.15) | 0.84 (0.15) | 0.84 (0.08) | **0.83** (0.07) | 0.83 (0.08) |
| MGraphDTA | 0.53 (0.59) | **0.57** (0.53) | **0.54** (0.50) | 0.86 (0.16) | **0.84** (0.16) | 0.85 (0.14) | 0.84 (0.08) | **0.83** (0.08) | **0.84** (0.08) |
| FDA | 0.53 (0.59) | 0.76 (0.68) | 1.30 (0.66) | 0.86 (0.16) | 0.83 (0.16) | 0.67 (0.17) | 0.84 (0.08) | **0.83** (0.08) | 0.75 (0.09) |
| Boltz-2 | 0.53 (0.60) | 0.76 (0.36) | 0.75 (0.26) | 0.86 (0.16) | 0.78 (0.13) | 0.79 (0.09) | 0.84 (0.08) | 0.80 (0.07) | 0.80 (0.06) |

suggest that AttentionDTA is particularly effective at adapting to ligand-induced variability. In contrast, the FDA and Boltz-2 are the models that does not benefit from fine-tuning; rather than improving, its performance stagnates or even deteriorates after incorporating the few-shot examples, suggesting a need for more effective fine-tuning strategies.

# 6 Limitation

Despite the addition of modified protein–ligand pairs, bringing the DAVIS dataset to 31,824 entries, its size remains limited for training data-intensive deep learning models. Its kinase-centric focus further restricts generalizability, as kinases represent only a fraction of the proteome. A more intrinsic limitation is the truncation of dissociation constants ($K_d$): roughly 70% of $K_d$ values are capped at $10\mu M$, obscuring weaker interactions and reducing data granularity. This censoring complicates the interpretation of modification-induced affinity changes, where $\Delta pK_d$ often represents only a lower bound or becomes entirely untrackable, thereby impairing predictive modeling. These limitations highlight the need for specialized algorithms and larger, more diverse datasets.

Table 5: Few-shot Modification Generalization benchmark. (a) Same-ligand, different-modifications: models are fine-tuned on limited modified protein–ligand pairs and evaluated on additional variants of the same kinase with a shared ligand. (b) Same-modification, different-ligands: models are fine-tuned and tested on distinct ligands targeting the same kinase modification. Metrics are mean (std) across kinase–ligand combinations.

| Model | MSE $\downarrow$ | | | | $R_P \uparrow$ | | | | C-index $\uparrow$ | | | |
|---|---|---|---|---|---|---|---|---|---|---|---|---|
| | $y_{WT}$ | $\hat{y}_{WT}$ | $\hat{y}$ | $\hat{y}_{FT}$ | $y_{WT}$ | $\hat{y}_{WT}$ | $\hat{y}$ | $\hat{y}_{FT}$ | $y_{WT}$ | $\hat{y}_{WT}$ | $\hat{y}$ | $\hat{y}_{FT}$ |
| (a) Same-ligand, different-modifications | | | | | | | | | | | | |
| DeepDTA | 0.64 (0.94) | 0.63 (0.75) | **0.62** (0.73) | **0.33** (0.43) | – | – | -0.06 (0.51) | 0.17 (0.52) | – | – | 0.46 (0.24) | 0.56 (0.23) |
| AttentionDTA | 0.64 (0.94) | 0.70 (0.94) | 0.68 (0.92) | 0.34 (0.45) | – | – | -0.03 (0.54) | 0.09 (0.61) | – | – | 0.47 (0.26) | 0.53 (0.28) |
| GraphDTA | 0.64 (0.94) | 0.71 (0.81) | 1.32 (2.09) | 0.83 (1.18) | – | – | -0.16 (0.68) | 0.02 (0.71) | – | – | 0.43 (0.33) | 0.50 (0.34) |
| DGraphDTA | 0.64 (0.94) | **0.62** (0.82) | 0.63 (0.81) | 0.37 (0.49) | – | – | -0.03 (0.46) | 0.05 (0.51) | – | – | 0.48 (0.21) | 0.52 (0.25) |
| MGraphDTA | 0.64 (0.94) | **0.62** (0.83) | **0.62** (0.82) | 0.43 (0.55) | – | – | -0.06 (0.46) | -0.04 (0.50) | – | – | 0.45 (0.23) | 0.48 (0.22) |
| FDA | 0.64 (0.94) | 0.87 (1.22) | 1.28 (1.80) | 0.36 (0.45) | – | – | **0.20** (0.75) | **0.21** (0.70) | – | – | **0.60** (0.35) | 0.56 (0.33) |
| Boltz-2 | 0.64 (0.94) | 0.79 (0.86) | 0.79 (0.92) | 0.42 (0.69) | – | – | 0.14 (0.61) | 0.20 (0.91) | – | – | 0.56 (0.29) | **0.58** (0.44) |
| (b) Same-modification, different-ligands | | | | | | | | | | | | |
| DeepDTA | 0.56 (0.87) | 0.63 (0.74) | 0.62 (0.67) | 0.54 (0.41) | 0.78 (0.28) | 0.76 (0.28) | 0.76 (0.26) | 0.78 (0.25) | 0.82 (0.14) | 0.80 (0.13) | 0.80 (0.13) | 0.81 (0.12) |
| AttentionDTA | 0.56 (0.87) | 0.67 (0.89) | 0.62 (0.75) | **0.42** (0.45) | 0.78 (0.28) | 0.77 (0.27) | 0.77 (0.26) | **0.80** (0.24) | 0.82 (0.14) | **0.81** (0.13) | **0.81** (0.13) | **0.82** (0.12) |
| GraphDTA | 0.56 (0.87) | 1.10 (1.09) | 1.45 (1.44) | 1.20 (1.18) | 0.78 (0.28) | 0.75 (0.26) | 0.66 (0.34) | 0.70 (0.29) | 0.82 (0.14) | 0.79 (0.12) | 0.76 (0.13) | 0.78 (0.11) |
| DGraphDTA | 0.56 (0.87) | 0.63 (0.82) | 0.64 (0.80) | 0.50 (0.61) | 0.78 (0.28) | **0.78** (0.28) | **0.78** (0.28) | 0.78 (0.28) | 0.82 (0.14) | **0.81** (0.13) | 0.80 (0.13) | 0.81 (0.13) |
| MGraphDTA | 0.56 (0.87) | **0.59** (0.79) | **0.55** (0.67) | 0.45 (0.39) | 0.78 (0.28) | 0.77 (0.29) | **0.78** (0.27) | **0.80** (0.24) | 0.82 (0.14) | 0.80 (0.14) | **0.81** (0.13) | **0.82** (0.12) |
| FDA | 0.56 (0.87) | 0.79 (1.08) | 1.15 (1.09) | 2.27 (1.62) | 0.78 (0.28) | 0.75 (0.27) | 0.67 (0.21) | 0.56 (0.31) | 0.82 (0.14) | 0.80 (0.12) | 0.74 (0.10) | 0.70 (0.12) |
| Boltz-2 | 0.56 (0.87) | 0.74 (0.61) | 0.70 (0.47) | 0.62 (0.48) | 0.78 (0.28) | 0.73 (0.24) | 0.76 (0.22) | 0.74 (0.28) | 0.82 (0.14) | 0.78 (0.11) | 0.79 (0.10) | 0.77 (0.14) |

Benchmarking docking-based approaches, such as Folding-Docking-Affinity (FDA) and Boltz-2, presents additional challenges. Although both demonstrate stronger zero-shot generalization than docking-free models, potential data leakage arises from overlaps between the curated DAVIS-complete dataset and the training data used for binding pose prediction (comprehensive analysis provided in Appendix H). Consequently, the distinct splits—'New-ligand,' 'New-protein,' and 'Both-new'—may not genuinely represent unseen data in this context, suggesting that performance metrics are likely overestimated. These overlaps underscore the necessity for entirely new benchmark datasets that strictly exclude previously seen proteins, ligands, and their combinations.

Furthermore, intuitively, one might expect the docking-based FDA model to outperform docking-free models completely, as it explicitly captures atom-level protein–ligand interactions, potentially reflecting the structural effects of protein modifications. Our structural investigations, however, suggest that the structure prediction models are not yet fully capable of capturing the structural variations introduced by modifications. For example, we observed that AlphaFold3 [4] predicts a phosphorylated state for both the non-phosphorylated and phosphorylated forms of the ABL1 protein, failing to distinguish between the two. The result is consistent with a recent study [33]. This underscores that subtle structural changes from protein modifications are not yet adequately captured by existing models, limiting the effectiveness of downstream binding affinity predictions.

# 7 Conclusion

Protein modifications significantly impact protein–ligand interactions and binding affinity, yet experimentally homogeneous datasets incorporating these modifications remain scarce. We address this gap by curating a complete version of the DAVIS dataset with previously ignored modified kinase proteins. Using three benchmarks—Augmented Dataset Prediction, Wild-Type to Modification Generalization, and Few-Shot Modification Generalization—we evaluate state-of-the-art models' abilities to distinguish protein modifications. Results indicate docking-based models demonstrate superior generalization in zero-shot scenarios. Conversely, docking-free models frequently overfit to wild-type proteins, encountering difficulty with unseen modifications; however, their performance improves notably after fine-tuning on a limited number of modified examples. This curated dataset and benchmarks offer valuable resources to advance generalizable affinity prediction models and precision medicine.

# 8 Author contributions

MH.W, Z.X, and D.Z conceived the research project. Z.X, S.J, and D.Z supervised the research project. MH.W developed the computational method, implemented the software, and performed the evaluation analyses. All authors analyzed the results and participated in the interpretation. MH.W wrote the manuscript with support from all other authors.

# 9 Acknowledgements

This work was not supported by any funding. We thank the creators of the original DAVIS dataset for their work and making their data publicly available.

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

# Appendix

## A   Details of Protein Modifications and Domain Annotations

Table A1: Summary of protein modifications and domain annotations in the DAVIS dataset

| Protein | Modification / Domain | Note |
|---|---|---|
| ABL1 | E255K-phosphorylated
F317I
F317I-phosphorylated
F317L
F317L-phosphorylated
H396P
H396P-phosphorylated
M351T-phosphorylated
Q252H
Q252H-phosphorylated
T315I
T315I-phosphorylated
Y253F-phosphorylated
Wild-type-phosphorylated | The phosphorylation site is Tyr393 [1].
–
The phosphorylation site is Tyr393 [1].
–
The phosphorylation site is Tyr393 [1].
–
The phosphorylation site is Tyr393 [1].
The phosphorylation site is Tyr393 [1].
–
The phosphorylation site is Tyr393 [1].
–
The phosphorylation site is Tyr393 [1].
The phosphorylation site is Tyr393 [1].
The phosphorylation site is Tyr393 [1]. |
| BRAF | V600E | – |
| CDK4 | CDK4-cyclinD1
CDK4-cyclinD3 | CDK4-cyclinD1 complex
CDK4-cyclinD3 complex |
| EGFR | E746A750del
G719C
G719S
L747E749del, A750P
L747E752del, P753S
L747E751del, Sins
L858R
L858R, T790M
L861Q
S752I759del
T790M | –
–
–
–
–
–
–
–
–
–
– |
| FGFR3 | G697C | – |
| FLT3 | D835H
D835Y
ITD
K663Q
N841I
R834Q | –
–
VDFREYEYDH insertion between Y591 and V592 [1]
–
–
– |
| GCN2 | Kinase domain 2, S808G | residues 590–1001 in UniProt Q9P2K8 |
| JAK1 | JH1 domain catalytic
JH2 domain pseudokinase | residues 875–1153 in UniProt P23458 [12]
residues 583–855 in UniProt P23458 [12] |
| JAK2
JAK3 | JH1 domain catalytic
JH1 domain catalytic | residues 849–1124 in UniProt O60674 [12]
residues 822–1111 in UniProt P52333 [12] |
| KIT | A829P
D816H
D816V
L576P
V559D
V559D, T670I
V559D, V654A | –
–
–
–
–
–
– |
| LRRK2 | G2019S | – |
| MET | M1250T | – |

| Protein | Modification | Note |
|---------|--------------|------|
| | Y1235D | – |
| PIK3CA | C420R | – |
| | E542K | – |
| | E545A | – |
| | E545K | – |
| | H1047L | – |
| | H1047Y | – |
| | I800L | – |
| | M1043I | – |
| | Q546K | – |
| RET | M918T | – |
| | V804L | – |
| | V804M | – |
| RPS6KA4 | Kinase domain 1 N-terminal | residues 33–301 in UniProt O75676 |
| | Kinase domain 2 C-terminal | residues 411–674 in UniProt O75676 |
| RPS6KA5 | Kinase domain 1 N-terminal | residues 49–318 in UniProt O75582 |
| | Kinase domain 2 C-terminal | residues 426–687 in UniProt O75582 |
| RSK1 | Kinase domain 1 N-terminal | residues 62–321 in UniProt Q15418 |
| | Kinase domain 2 C-terminal | residues 418–675 in UniProt Q15418 |
| RSK2 | Kinase domain 1 N-terminal | residues 68–327 in UniProt P51812 |
| RSK3 | Kinase domain 1 N-terminal | residues 59–318 in UniProt Q15349 |
| | Kinase domain 2 C-terminal | residues 415–672 in UniProt Q15349 |
| RSK4 | Kinase domain 1 N-terminal | residues 73–330 in UniProt Q9UK32 |
| | Kinase domain 2 C-terminal | residues 426–683 in UniProt Q9UK32 |
| TYK2 | JH1 domain catalytic | residues 897–1176 in UniProt P29597 [12] |
| | JH2 domain pseudokinase | residues 589–875 in UniProt P29597 [12] |

# B   Binding Affinity Distribution

The DAVIS dataset contains 31,824 protein–ligand binding affinity measurements (442 proteins $\times$ 72 ligands). It is well known for its pronounced imbalance in binding affinity distribution: approximately 70% of protein–ligand pairs have $K_d$ values capped at 10 $\mu$M ($pK_d$ set to 5), leading to an overrepresentation of lower-affinity interactions [10]. The uncapped binding affinity distribution for all pairs is shown in Fig. A1(a). Among these, wild-type protein–ligand pairs include 20,126 capped measurements, with the corresponding uncapped distribution shown in Fig. A1(b). Modified protein–ligand pairs include 2,274 capped measurements, with the uncapped distribution shown in Fig. A1(c).

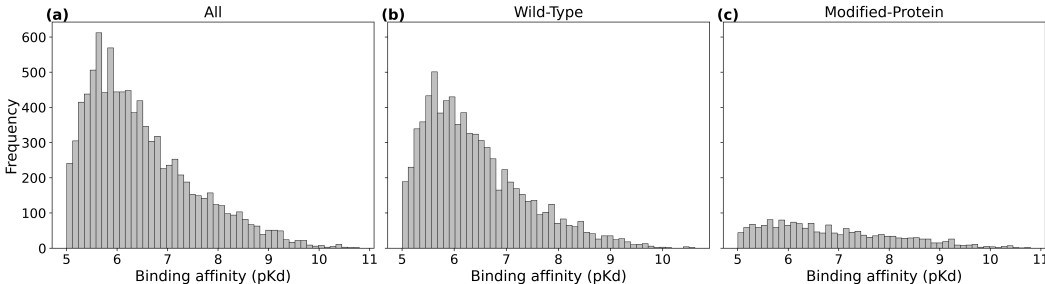

Figure A1: Distribution of uncapped binding affinity ($pK_d$) for (a) all protein–ligand pairs, (b) wild-type proteins, and (c) modified proteins

# C   Binding Affinity Alternation

To systematically evaluate how different protein modifications affect binding affinity, we performed a quantitative analysis and visualized the results as a heatmap (Fig. A2). The heatmap shows the magnitude of binding affinity

changes induced by modifications across ABL1, BRAF, EGFR, FGFR3, FLT3, KIT, LRRK2, MET, PIK3CA, and RET. GCN2 has modified variants in the dataset, but its wild-type form is missing; therefore, affinity changes for GCN2 are not calculable and are excluded. The affinity change is defined as

$$\Delta pK_d = A(p_j^{m_i}, l_k) - A(p^{w_i}, l_k)$$

A key limitation of the DAVIS dataset is that binding affinity measurements ($K_d$) are capped at values above $10\mu$M. As a result, we categorized modification-induced changes in binding affinity into four groups, as summarized in Table. A2 (1) WT-uncapped & modification-uncapped: both wild-type and modified proteins have $K_d$ values below $10\mu$M. The affinity changes are precisely trackable, and $\Delta pK_d$ reflects the exact magnitude of change. The distribution of these changes is shown in Fig. A3(a) (2) WT-capped & modification-uncapped: wild-type is capped ($K_d > 10 \ \mu$M), while the modified protein is not. This indicates an increase in affinity due to the modification. However, since the wild-type value is unknown beyond the threshold, $\Delta pK_d$ only represents a lower bound of the actual change. The distribution is shown in Fig. A3(b) (3) WT-uncapped & modification-capped: modified protein is capped, while the wild-type is not. This suggests a decrease in affinity, but again, $\Delta pK_d$ captures only the minimum possible magnitude, making the true change untrackable. The distribution is shown in Fig. A3(c) (4) WT-capped & modification-capped: both wild-type and modified proteins have capped affinities. In this case, the exact change in binding affinity is completely untrackable, and $\Delta pK_d = 0$

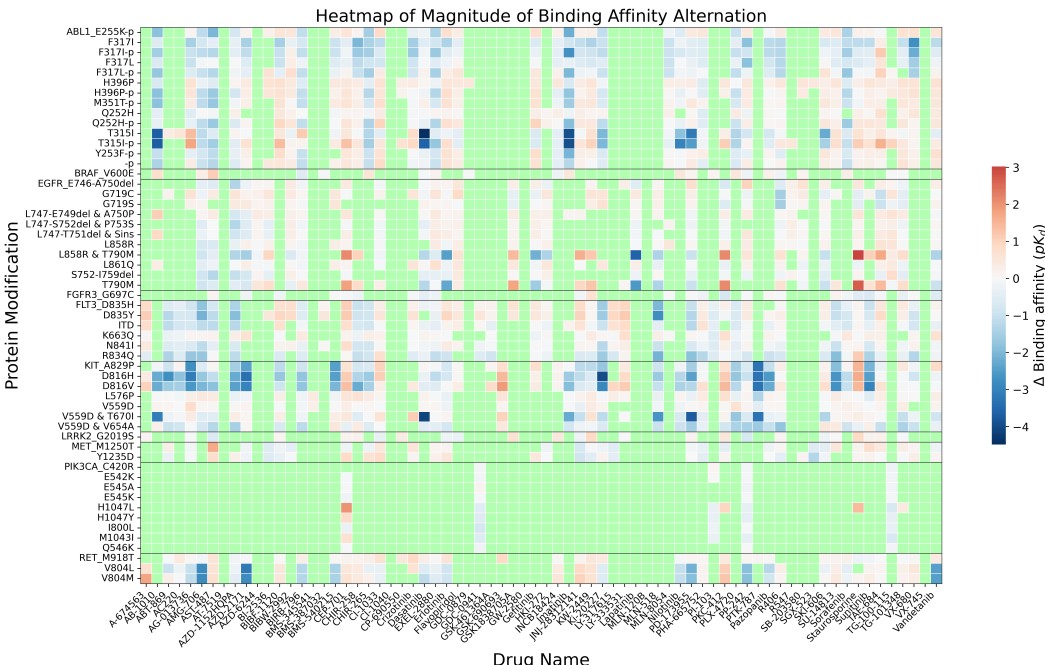

Figure A2: Heatmap of magnitude of binding affinity change. Colors from blue to red represent either the exact magnitude or the lower bound of the change, while light green indicates untrackable changes.

Table A2: Binding affinity summary between wild-type and modified proteins. ✓ indicates a protein-ligand pair with $K_d < 10 \ \mu$M, while ✗ indicates $K_d > 10 \ \mu$M.

| WT | Modification | # |
|----|--------------|-----|
| ✓ | ✓ | 1601 |
| ✗ | ✓ | 134 |
| ✓ | ✗ | 157 |
| ✗ | ✗ | 2068 |

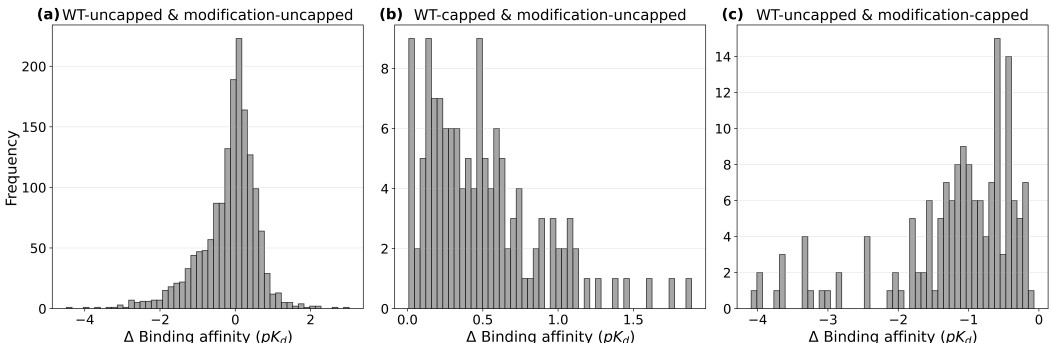

Figure A3: Distribution of magnitude of trackable binding affinity alternation. (a) Both wild-type and modified proteins have $K_d$ values below $10\mu$M. The affinity changes are precisely trackable. (b) Wild-type is capped ($K_d > 10\ \mu$M), while the modified protein is not. (c) Modified protein is capped, while the wild-type is not.

## D   Input preprocessing

For the all benchmarks, the protein kinase domain is exclusively selected for each kinase protein. If domain annotations are absent in the DAVIS dataset, we utilize domain information from the UniProt database [1]. Phosphorylation events are not accounted for in docking-free methods due to their intrinsic input constraints; however, phosphorylated protein structures used as inputs for the FDA model are predicted using AlphaFold3 [4]. Similarly, AlphaFold3 is employed to predict structures of other modified proteins, whereas wild-type protein structures are directly sourced from the AlphaFold Protein Structure Database [6]. For the CDK4-cyclinD1 and CDK4-cyclinD3 complexes, amino acid sequences of both components are concatenated for docking-free model inputs, whereas their 3D structures are predicted using AlphaFold3 for the FDA model.

## E   Benchmark Models

We include 7 models (5 docking free-based models and 2 docking-based model) in the benchmarks.

**DeepDTA**   DeepDTA [55] takes drug SMILES strings and protein amino acid sequences as input. It uses two parallel 1D CNNs to extract features from the drug and protein sequences, respectively. These features are then concatenated and passed through fully connected layers to predict the binding affinity. Our implementation is based on the code available at `https://github.com/KSUN63/DeepDTA-Pytorch`.

**AttentionDTA**   AttentionDTA [54] takes SMILES strings and protein sequences as input but enhances DeepDTA by adding attention mechanisms. After initial feature extraction using 1D CNNs for both inputs, multi-head attention layers are applied to focus on important regions in the sequences. Our implementation is based on the code available at `https://github.com/zhaoqichang/AttentionDTA_TCBB`.

**GraphDTA**   GraphDTA [27] represents the drug as a molecular graph (with atoms as nodes and bonds as edges, derived from the SMILES) and the protein as a sequence. A graph neural network (GCN and GAT) is used to process the drug graph, while a CNN processes the protein sequence. The learned representations are concatenated and passed to fully connected layers for affinity prediction. Our implementation is based on the code available at `https://github.com/thinng/GraphDTA`.

**DGraphDTA**   DGraphDTA [15] encodes both the drug and the protein as graphs: drugs from molecular structures (via SMILES) and proteins from predicted contact maps. It applies GCNs to each graph separately, then combines the learned representations to predict the binding affinity. Our implementation is based on the code available at `https://github.com/595693085/DGraphDTA`.

**MGraphDTA**   MGraphDTA [50] processes drug molecules as graphs and proteins as amino acid sequences. It employs a deep, multiscale graph neural network architecture with stacked GNN layers and dense skip connections to capture hierarchical structural features of the drug. Protein sequences are processed via 1D CNNs.

The fused representations are used to predict binding affinity. Our implementation is based on the code available at `https://github.com/guaguabujianle/MGraphDTA`.

**Folding-Docking-Affinity** Folding-Docking-Affinity [47] uses protein sequences and drug molecular SMILES as input. The model operates in three stages: first, it predicts the 3D structure of the protein (e.g., via AlphaFold [4, 11]); second, it docks the drug molecule onto the predicted protein structure to generate a 3D complex through DiffDock [7]; third, it uses the 3D conformation of the protein-ligand complex as input to predict binding affinity through GIGN [51]. Notably, in our implementation, we ensemble five affinity predictors, and the final output is the mean of their predictions. Our implementation is based on the code available at `https://github.com/ZhiGroup/FDA`.

**Boltz-2** Boltz-2 [32] is an open-source biomolecular foundation model that jointly predicts protein–ligand complex structures and binding affinities. During structure generation we kept the Trunk module frozen and applied the default affinity-inference configuration (5 samples, 200 reverse-diffusion steps per sample). Complexes were ranked by the protein–ligand pair ipTM score; the top-ranked pose was subsequently used to retrain the affinity module (4 Pairformer layers) from scratch on the curated DAVIS dataset. Our implementation is based on the code available at `https://github.com/AustinApple/boltz/tree/train_affinity_module`.

# F   Model Training Details

Table A3: Training hyperparameter settings for docking free-based models, Folding-Docking-Affinity, and Boltz-2 used in Augmented Dataset Prediction and Wild-Type to Modification Generalization benchmarks.

| Hyperparameter | Docking free-based models | Folding-Docking-Affinity | Boltz-2 |
|---|---|---|---|
| Optimizer | Adam | Adam | AdamW |
| Learning rate | 5e-4 | 5e-4 | 3e-4 |
| Weight decay | – | 1e-6 | – |
| Batch size | 64 | 128 | 64 |
| Max epochs | 1,000 | 1,000 | 100 |
| Early stopping patience | 100 | 100 | 5 |

Table A4: Fine-tuning hyperparameter settings for docking free-based models, Folding-Docking-Affinity, and Boltz-2 used in Few-Shot Modification Generalization.

| Hyperparameter | Docking free-based models | Folding-Docking-Affinity | Boltz-2 |
|---|---|---|---|
| **Same-ligand, different-modifications** | | | |
| Optimizer | Adam | Adam | AdamW |
| Learning rate | 5e-4 | 5e-3 | 3e-4 |
| Weight decay | – | 1e-6 | – |
| Batch size | len(training set) | len(training set) | len(training set) |
| Number of epochs | 30 | 30 | 10 |
| **Same-modification, different-ligands** | | | |
| Optimizer | Adam | Adam | AdamW |
| Learning rate | 1e-4 | 1e-4 | 3e-4 |
| Weight decay | – | 1e-6 | – |
| Batch size | len(training set) | len(training set) | len(training set) |
| Number of epochs | 10 | 10 | 10 |

# G   Metrics for Model Evaluation

We introduce two baseline methods for the same-ligand, different-modifications and same-modification, different-ligands settings: Wild-type ground truth ($y_{WT}$), which uses the ground truth binding affinity of the wild-type

protein–ligand pair to predict that of the modified pairs, and wild-type prediction ($\hat{y}_{\text{WT}}$), which uses the model-predicted affinity of the wild-type pair instead. We use both baselines to compute evaluation metrics—MSE, $R_p$, and C-index—for comparison with the predictions on the modified pairs. The following is the calculation details:

## G.1 Same-ligand, different-modifications

$$\text{MSE}(y, y_{\text{WT}}) = \frac{1}{n} \sum_{j=1}^{n} (A(p^{w_i}, l_k) - A(p_j^{m_i}, l_k))^2$$

where the ground-truth binding affinity of the wild-type kinase–ligand pair, $A(p^{w_i}, l_k)$, is used to compute Mean Squared Error, denoted as $\text{MSE}(y, y_{\text{WT}})$.

$$\text{MSE}(y, \hat{y}_{\text{WT}}) = \frac{1}{n} \sum_{j=1}^{n} (f(p^{w_i}, l_k) - A(p_j^{m_i}, l_k))^2$$

where the model-predicted binding affinity for the wild-type kinase–ligand pair, $f(p^{w_i}, l_k)$, is used to compute Mean Squared Error, denoted as $\text{MSE}(y, \hat{y}_{\text{WT}})$.

$$\text{MSE} = \frac{1}{n} \sum_{j=1}^{n} (f(p_j^{m_i}, l_k) - A(p_j^{m_i}, l_k))^2$$

where the model-predicted binding affinity for the modified kinase–ligand pairs, $f(p_j^{m_i}, l_k)$, is used to compute Mean Squared Error, denoted as MSE.

$$R_p = \frac{\sum_{j=1}^{n} (f(p_j^{m_i}, l_k) - \overline{f(p_j^{m_i}, l_k)})(A(p_j^{m_i}, l_k) - \overline{A(p_j^{m_i}, l_k)})}{\sqrt{\sum_{j=1}^{n} (f(p_j^{m_i}, l_k) - \overline{f(p_j^{m_i}, l_k)})^2} \sqrt{\sum_{j=1}^{n} (A(p_j^{m_i}, l_k) - \overline{A(p_j^{m_i}, l_k)})^2}}$$

where the model-predicted binding affinity, $f(p_j^{m_i}, l_k)$, and ground-truth binding affinity, $A(p_j^{m_i}, l_k)$, for the modified kinase–ligand pairs are used to compute Pearson correlation coefficient, $R_p$.

$$\text{C-index} = \frac{\sum_{r \neq s} I(A(p_r^{m_i}, l_k) > A(p_s^{m_i}, l_k)) \cdot I(f(p_r^{m_i}, l_k) > f(p_s^{m_i}, l_k)) + 0.5 \cdot I(f(p_r^{m_i}, l_k) = f(p_s^{m_i}, l_k))}{\sum_{r \neq s} I(A(p_r^{m_i}, l_k) > A(p_s^{m_i}, l_k))}$$

where the model-predicted binding affinity, $f(p_{r/s}^{m_i}, l_k)$, and ground-truth binding affinity, $A(p_{r/s}^{m_i}, l_k)$, for the modified kinase–ligand pairs are used to compute C-index. $I(\cdot)$ is denoted as an indicator function that returns 1 if the condition is true and 0 otherwise

## G.2 Same-modification, different-ligands

$$\text{MSE}(y, y_{\text{WT}}) = \frac{1}{n} \sum_{k=1}^{n} (A(p^{w_i}, l_k) - A(p_j^{m_i}, l_k))^2$$

where the ground-truth binding affinity of the wild-type kinase–ligand pairs, $A(p^{w_i}, l_k)$, is used to compute Mean Squared Error.

$$\text{MSE}(y, \hat{y}_{\text{WT}}) = \frac{1}{n} \sum_{k=1}^{n} (f(p^{w_i}, l_k) - A(p_j^{m_i}, l_k))^2$$

where the model-predicted binding affinity for the wild-type kinase–ligand pairs, $f(p^{w_i}, l_k)$, is used to compute Mean Squared Error.

$$\text{MSE} = \frac{1}{n} \sum_{k=1}^{n} (f(p_j^{m_i}, l_k) - A(p_j^{m_i}, l_k))^2$$

where the model-predicted binding affinity for the modified kinase–ligand pairs, $f(p_j^{m_i}, l_k)$, is used to compute Mean Squared Error.

$$R_p(y, y_{\text{WT}}) = \frac{\sum_{k=1}^{n}(A(p^{w_i}, l_k) - \overline{A(p^{w_i}, l_k)})(A(p_j^{m_i}, l_k) - \overline{A(p_j^{m_i}, l_k)})}{\sqrt{\sum_{k=1}^{n}(A(p^{w_i}, l_k) - \overline{A(p^{w_i}, l_k)})^2}\sqrt{\sum_{k=1}^{n}(A(p_j^{m_i}, l_k) - \overline{A(p_j^{m_i}, l_k)})^2}}$$

where the ground-truth binding affinity, $A(p^{w_i}, l_k)$, for the wild-type kinase–ligand pairs and the ground-truth binding affinity, $A(p_j^{m_i}, l_k)$, for the modified kinase–ligand pairs are used to compute Pearson correlation coefficient.

$$R_p(y, \hat{y}_{\text{WT}}) = \frac{\sum_{k=1}^{n}(f(p^{w_i}, l_k) - \overline{f(p^{w_i}, l_k)})(A(p_j^{m_i}, l_k) - \overline{A(p_j^{m_i}, l_k)})}{\sqrt{\sum_{k=1}^{n}(f(p^{w_i}, l_k) - \overline{f(p^{w_i}, l_k)})^2}\sqrt{\sum_{k=1}^{n}(A(p_j^{m_i}, l_k) - \overline{A(p_j^{m_i}, l_k)})^2}}$$

where the model-predicted binding affinity, $f(p^{w_i}, l_k)$, for the wild-type kinase–ligand pairs and the ground-truth binding affinity, $A(p_j^{m_i}, l_k)$, for the modified kinase–ligand pairs are used to compute Pearson correlation coefficient.

$$R_p = \frac{\sum_{k=1}^{n}(f(p_j^{m_i}, l_k) - \overline{f(p_j^{m_i}, l_k)})(A(p_j^{m_i}, l_k) - \overline{A(p_j^{m_i}, l_k)})}{\sqrt{\sum_{k=1}^{n}(f(p_j^{m_i}, l_k) - \overline{f(p_j^{m_i}, l_k)})^2}\sqrt{\sum_{k=1}^{n}(A(p_j^{m_i}, l_k) - \overline{A(p_j^{m_i}, l_k)})^2}}$$

where the model-predicted binding affinity, $f(p_j^{m_i}, l_k)$, and ground-truth binding affinity, $A(p_j^{m_i}, l_k)$, for the modified kinase–ligand pairs are used to compute Pearson correlation coefficient, $R_p$.

$$\text{C-index}(y, y_{\text{WT}}) = \frac{\sum_{r \neq s} I(A(p_j^{m_i}, l_r) > A(p_j^{m_i}, l_s)) \cdot I(A(p^{w_i}, l_r) > A(p^{w_i}, l_s)) + 0.5 \cdot I(A(p^{w_i}, l_r) = A(p^{w_i}, l_s))}{\sum_{r \neq s} I(A(p_j^{m_i}, l_r) > A(p_j^{m_i}, l_s))}$$

where the ground-truth binding affinity, $A(p^{w_i}, l_{r/s})$, for the wild-type kinase–ligand pairs and the ground-truth binding affinity, $A(p_j^{m_i}, l_{r/s})$, for the modified kinase–ligand pairs are used to compute concordance index.

$$\text{C-index}(y, \hat{y}_{\text{WT}}) = \frac{\sum_{r \neq s} I(A(p_j^{m_i}, l_r) > A(p_j^{m_i}, l_s)) \cdot I(f(p^{w_i}, l_r) > f(p^{w_i}, l_s)) + 0.5 \cdot I(f(p^{w_i}, l_r) = f(p^{w_i}, l_s))}{\sum_{r \neq s} I(A(p_j^{m_i}, l_r) > A(p_j^{m_i}, l_s))}$$

where the model-predicted binding affinity, $f(p^{w_i}, l_{r/s})$, for the wild-type kinase–ligand pairs and the ground-truth binding affinity, $A(p_j^{m_i}, l_{r/s})$, for the modified kinase–ligand pairs are used to compute concordance index.

$$\text{C-index} = \frac{\sum_{r \neq s} I(A(p_j^{m_i}, l_r) > A(p_j^{m_i}, l_s)) \cdot I(f(p_j^{m_i}, l_r) > f(p_j^{m_i}, l_s)) + 0.5 \cdot I(f(p_j^{m_i}, l_r) = f(p_j^{m_i}, l_s))}{\sum_{r \neq s} I(A(p_j^{m_i}, l_r) > A(p_j^{m_i}, l_s))}$$

where the model-predicted binding affinity, $f(p_j^{m_i}, l_{r/s})$, and ground-truth binding affinity, $A(p_j^{m_i}, l_{r/s})$, for the modified kinase–ligand pairs are used to compute concordance index.

## H  Assessment of Data Leakage

While docking-based models, such as Folding-Docking-Affinity (FDA) and Boltz-2, exhibit superior zero-shot generalization compared to docking-free baselines, this performance may be partially attributable to data leakage. Specifically, we identified overlaps between the DAVIS-complete benchmark and the training corpora used for binding pose prediction. A quantitative breakdown of these overlaps is presented below:

Table A5: Analysis of data overlap between the DAVIS-complete dataset and the training sources of AlphaFold-Multimer (folding component of FDA) and DiffDock (Docking component of FDA). Overlap is defined as 100% sequence identity for proteins and a Tanimoto coefficient of 1.0 (Morgan fingerprints) for ligands. A protein-ligand pair is considered overlapping only if both components meet these criteria. Values indicate the number of shared entities relative to the total count in DAVIS-complete. Numbers in parentheses represent the subset of modified proteins and their corresponding pairs.

| | Alphafold-Multimer (Folding) | DiffDock (Docking) |
|---|---|---|
| protein (modified protein) | 212/444 (9/56) | 144/444 (9/56) |
| ligand | – | 36/72 |
| protein-ligand (modified protein-ligand) | – | 77/31,824 (3/4,032) |

Table A6: Analysis of data overlap between the DAVIS-complete dataset and Boltz-2 structural prediction training sources (PDB, MISATO, ATLAS, and mdCATH). Overlap is defined as 100% sequence identity for proteins and a Tanimoto coefficient of 1.0 (Morgan fingerprints) for ligands. A protein-ligand pair is considered overlapping only if both components meet these criteria. Values indicate the number of shared entities relative to the total count in DAVIS-complete. Numbers in parentheses represent the subset of modified proteins and their corresponding pairs.

|  | PDB | MISATO | ATLAS | mdCATH |
|---|---|---|---|---|
| protein (modified protein) | 246/444 (13/56) | 145/444 (9/56) | 0/444 (0/56) | 4/444 (0/56) |
| ligand | 47/72 | 36/72 | – | – |
| protein-ligand (modified protein-ligand) | 119/31,824 (6/4,032) | 80/31,824 (3/4,032) | – | – |

# I Supplementary figures

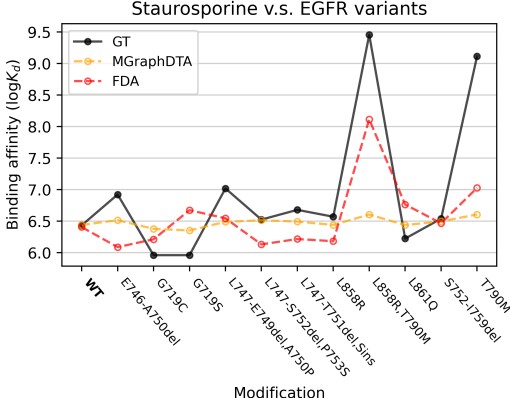

Figure A4: A case of the Wild-Type to Modification Generalization benchmark: Same-ligand, different-modifications — Staurosporine binding to various EGFR protein variants.

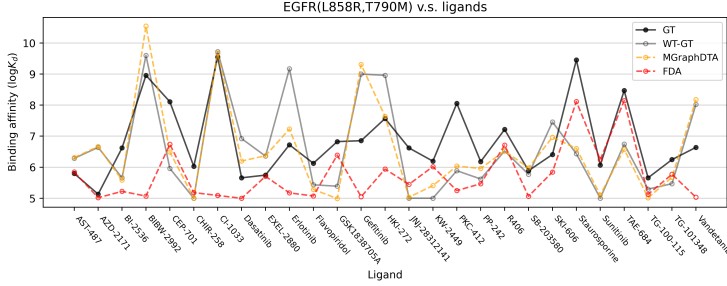

Figure A5: A case of the Wild-Type to Modification Generalization benchmark: Same-modification, different-ligands — the EGFR(L858R, T790M) variant interacting with multiple ligands.

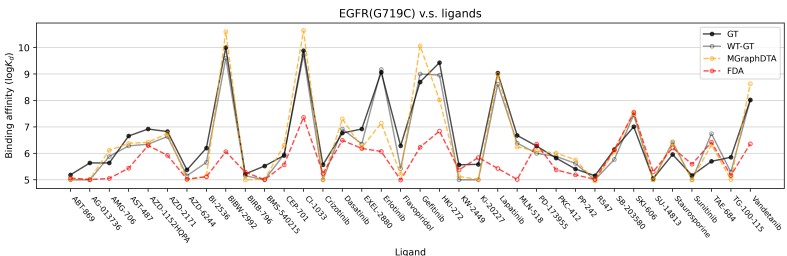

Figure A6: A case of the Wild-Type to Modification Generalization benchmark: Same-modification, different-ligands — the EGFR(G719C) variant interacting with multiple ligands.

## Appendix References

[1] Lisa M Wodicka, Pietro Ciceri, Mindy I Davis, Jeremy P Hunt, Mark Floyd, Sara Salerno, Xuequn H Hua, Julia M Ford, Robert C Armstrong, Patrick P Zarrinkar, et al. Activation state-dependent binding of small molecule kinase inhibitors: structural insights from biochemistry. *Chemistry & biology*, 17(11):1241–1249, 2010.

[2] Xiaoyi Hu, Jing Li, Maorong Fu, Xia Zhao, and Wei Wang. The jak/stat signaling pathway: from bench to clinic. *Signal transduction and targeted therapy*, 6(1):402, 2021.

[3] Mindy I. Davis, Jeremy P. Hunt, Sanna Herrgard, Pietro Ciceri, Lisa M. Wodicka, Gabriel Pallares, Michael Hocker, Daniel K. Treiber, and Patrick P. Zarrinkar. Comprehensive analysis of kinase inhibitor selectivity. *Nature Biotechnology*, 29:1046–1051, 11 2011.

[4] Uniprot: the universal protein knowledgebase in 2025. *Nucleic Acids Research*, 53(D1):D609–D617, 2025.

[5] Josh Abramson, Jonas Adler, Jack Dunger, Richard Evans, Tim Green, Alexander Pritzel, Olaf Ronneberger, Lindsay Willmore, Andrew J Ballard, Joshua Bambrick, et al. Accurate structure prediction of biomolecular interactions with alphafold 3. *Nature*, pages 1–3, 2024.

[6] Mihaly Varadi, Damian Bertoni, Paulyna Magana, Urmila Paramval, Ivanna Pidruchna, Malarvizhi Radhakrishnan, Maxim Tsenkov, Sreenath Nair, Milot Mirdita, Jingi Yeo, et al. Alphafold protein structure database in 2024: providing structure coverage for over 214 million protein sequences. *Nucleic acids research*, 52(D1):D368–D375, 2024.

[7] Hakime Öztürk, Arzucan Özgür, and Elif Ozkirimli. Deepdta: Deep drug-target binding affinity prediction. *Bioinformatics*, 34:i821–i829, 9 2018.

[8] Qichang Zhao, Guihua Duan, Mengyun Yang, Zhongjian Cheng, Yaohang Li, and Jianxin Wang. Attentiondta: Drug–target binding affinity prediction by sequence-based deep learning with attention mechanism. *IEEE/ACM transactions on computational biology and bioinformatics*, 20(2):852–863, 2022.

[9] Thin Nguyen, Hang Le, Thomas P Quinn, Tri Nguyen, Thuc Duy Le, and Svetha Venkatesh. Graphdta: predicting drug–target binding affinity with graph neural networks. *Bioinformatics*, 37(8):1140–1147, 2021.

[10] Mingjian Jiang, Zhen Li, Shugang Zhang, Shuang Wang, Xiaofeng Wang, Qing Yuan, and Zhiqiang Wei. Drug–target affinity prediction using graph neural network and contact maps. *RSC advances*, 10(35):20701–20712, 2020.

[11] Ziduo Yang, Weihe Zhong, Lu Zhao, and Calvin Yu-Chian Chen. Mgraphdta: deep multiscale graph neural network for explainable drug–target binding affinity prediction. *Chemical science*, 13(3):816–833, 2022.

[12] Ming-Hsiu Wu, Ziqian Xie, and Degui Zhi. A folding-docking-affinity framework for protein-ligand binding affinity prediction. *Communications Chemistry*, 8(1):1–9, 2025.

[13] Richard Evans, Michael O'Neill, Alexander Pritzel, Natasha Antropova, Andrew Senior, Tim Green, Augustin Žídek, Russ Bates, Sam Blackwell, Jason Yim, et al. Protein complex prediction with alphafold-multimer. *biorxiv*, pages 2021–10, 2021.

[14] Gabriele Corso, Hannes Stärk, Bowen Jing, Regina Barzilay, and Tommi Jaakkola. Diffdock: Diffusion steps, twists, and turns for molecular docking. *arXiv preprint arXiv:2210.01776*, 2022.

[15] Ziduo Yang, Weihe Zhong, Qiujie Lv, Tiejun Dong, and Calvin Yu-Chian Chen. Geometric interaction graph neural network for predicting protein-ligand binding affinities from 3d structures (gign). *The Journal of Physical Chemistry Letters*, 14(8):2020–2033, 2023.

[16] Saro Passaro, Gabriele Corso, Jeremy Wohlwend, Mateo Reveiz, Stephan Thaler, Vignesh Ram Somnath, Noah Getz, Tally Portnoi, Julien Roy, Hannes Stark, et al. Boltz-2: Towards accurate and efficient binding affinity prediction. *BioRxiv*, pages 2025–06, 2025.

