# OpenReview forum: "Towards precision protein-ligand affinity prediction benchmark: A Complete and Modification-Aware DAVIS Dataset"
_NeurIPS.cc/2025/Datasets_and_Benchmarks_Track — NeurIPS 2025 Datasets and Benchmarks Track poster_

### Official Review · Reviewer_2Jxd · 2025-06-19

**Rating:** 5
**Confidence:** 3

**Summary:**

To design therapies, we want to predict the binding of proteins and ligands. One current dataset used for validation is DAVIS, which is a collection of diverse protein ligand pairs with experimentally measured binding affinities. The authors here curate an extension of DAVIS to include the binding of slightly modified proteins. This allows them to benchmark models not only on binding prediction across diverse sequences, but also the predictions of the effects of small mutations.

**Dataset Code Accessibility:**

Yes

**Dataset Code Comments:**

The data is easily available on Huggingface.

**Ethical Considerations:**

No, there are no or only very minor ethics concerns

**Final Justification:**

Strong paper. It was also very straightforward, hence my shorted review. My concerns were well addressed in the rebuttal.

**Limitations Weaknesses:**

1. The authors do not describe the magnitude of the effects of mutations or their bias. Were modifications in the original database chosen according to some criterion? Did most of these modifications have a large impact on binding? How does this affect our interpretation of this metric?

**Strengths Contributions:**

1. The authors describe a simple extension to an established dataset, making adoption easy.

2. The authors explicitly connect their train-test split settings to realistic desiderata, such as predicting the effects of modifications.

3. The authors describe the dataset in some mathematical rigor in section 3, enhancing clarity.

4. The authors benchmark a number of models on these splits.

---

> ### Author Rebuttal · Authors · 2025-07-30
>
> We appreciate the reviewer’s thorough evaluation and thoughtful questions. Below, we provide our detailed responses to each point raised.
>
> **The authors do not describe the magnitude of the effects of mutations or their bias.**
>
> We appreciate the reviewer’s insightful comment and fully agree with its importance. To address this, we conducted a systematic and quantitative evaluation of the effects of modifications. The observed magnitude of binding affinity (pKd) alterations ranges from approximately –4.49 to +3.02. However, as noted in **Section 6**, a key limitation of the DAVIS dataset is that binding affinity measurements (Kd) are capped at values greater than 10 µM. As a result, for roughly half of the modified protein-ligand pairs—where both the wild-type and the modified proteins exhibit affinities beyond this threshold—we lose track of the exact magnitude of modification-induced changes. For the trackable alternations (ΔpKd ≠ 0), we summarize the distribution of pKd changes as the following:
>
> |        | Mean  | Std   | Min   | 25%   | 50%   | 75%   | Max   |
> |--------|-------|-------|-------|-------|-------|-------|-------|
> | Value  | -0.21 | 0.84  | -4.49 | -0.59 | -0.04 | 0.30  | 3.02  |
>
> This analysis shows that while many modifications have minor or negligible effects, a considerable number lead to increases or decreases in binding affinity. We will include this summary in the revised version to enhance clarity regarding the range and impact of modification-induced changes.
>
> **Were modifications in the original database chosen according to some criterion?**
>
> Yes, modifications included in the original dataset were selected based on criteria outlined by Davis et al [1]. Specifically, kinase modifications were chosen primarily based on their:
> - Disease relevance: Variants known to be linked to clinical outcomes or therapeutic resistance (e.g., EGFR T790M, RET V804L/M).
> - Distinct kinase conformational characteristics: Modifications exhibiting different kinase activation states or conformations (such as phosphorylated vs. non-phosphorylated ABL1 variants) were deliberately included to evaluate inhibitor selectivity concerning Type I and Type II binding modes.
>
> [1] Davis, M. I., Hunt, J. P., Herrgard, S., Ciceri, P., Wodicka, L. M., Pallares, G., ... & Zarrinkar, P. P. (2011). Comprehensive analysis of kinase inhibitor selectivity. Nature biotechnology, 29(11), 1046-1051.
>
> **Did most of these modifications have a large impact on binding? How does this affect our interpretation of this metric?**
>
> As noted above, the observed ΔpKd values range from approximately –4.49 to +3.02, while about half of the modified protein–ligand pairs exhibit untrackable changes in binding affinity due to dataset limitations. For these pairs, where both the wild-type and modified proteins exceed the measurable threshold, a model could achieve high accuracy by simply reproducing the wild-type affinity. This may lead to an overestimation of the model’s ability to generalize to cases involving genuine, modification-induced affinity shifts. To better understand this impact, we computed performance metrics separately for the trackable and untrackable subsets in the **Global Modification Generalization** benchmark. The results are shown in the table below.
> | Model        | Trackable                        |                          |                          | Untrackable                      |                          |                          |
> |--------------|----------------------------------|--------------------------|--------------------------|----------------------------------|--------------------------|--------------------------|
> |              | MSE                              | Rp                       | C-index                  | MSE                              | Rp                       | C-index                  |
> | DeepDTA      | 0.71 (0.00)                      | 0.80 (0.01)              | 0.79 (0.00)              | 0.01 (0.01)                      | -                        | -                        |
> | AttentionDTA | 0.72 (0.04)                      | 0.80 (0.01)              | 0.79 (0.00)              | 0.01 (0.00)                      | -                        | -                        |
> | GraphDTA     | 1.20 (0.05)                      | 0.66 (0.01)              | 0.73 (0.01)              | 0.05 (0.02)                      | -                        | -                        |
> | DGaphDTA     | 0.75 (0.04)                      | 0.80 (0.01)              | 0.79 (0.00)              | 0.01 (0.00)                      | -                        | -                        |
> | MGraphDTA    | 0.71 (0.02)                      | 0.80 (0.00)              | 0.79 (0.00)              | 0.01 (0.01)                      | -                        | -                        |
> | FDA          | 1.26 (0.01)                      | 0.65 (0.01)              | 0.73 (0.00)              | 0.05 (0.00)                      | -                        | -                        |
>
> As expected, the majority of prediction error originates from the trackable subset, while the untrackable subset is predicted with near-perfect accuracy. The near-perfect prediction on untrackable data dilutes the error from the trackable subset, leading to an inflated overall performance, shown in the **Table 3**. We will include this analysis and its implications in the revised version to help provide a clearer understanding of the benchmark results and ensure a more accurate interpretation of the model’s generalization performance.
>
>
> We appreciate the reviewer’s thoughtful feedback again and hope our clarifications address the concerns raised.

---

> > ### Comment · Reviewer_2Jxd · 2025-08-01
> > **response**
> >
> > This addresses my concerns. The dataset has biases and idiosyncrasies, like every other realistic dataset, but this documentation shows that these don't make it impossible to use these dataset as a benchmark.

---

### Official Review · Reviewer_bWgu · 2025-06-29

**Rating:** 4
**Confidence:** 4

**Summary:**

This paper constructs DAVIS-complete, an extended version of the DAVIS dataset that further incorporates protein modifications. Based on this enhanced dataset, three benchmarks are designed to evaluate model robustness in predicting binding affinities for modified proteins, including Augmented Dataset Prediction, Wild-Type to Modification Generalization, and Few-Shot Modification Generalization. Comparasions are conducted across five docking-free methods and a docking-based model (FDA), offering a comprehensive analysis of model performance under diverse scenarios.

**Dataset Code Accessibility:**

Yes

**Dataset Code Comments:**

The codes and datasets are provided with Croissant file attached.

**Ethical Considerations:**

No, there are no or only very minor ethics concerns

**Final Justification:**

My original concerns have been addressed by the authors, and I will maintain my original score.

**Limitations Weaknesses:**

1. To enable docking-free, sequence-based models to handle modified residues, the original vocabulary likely requires adaptation. For example, it should be clarified whether modified amino acids (e.g., phosphotyrosine) are treated as distinct tokens from their unmodified counterparts (e.g., tyrosine). A more detailed description of how modifications are represented in input sequences would strengthen the paper.

2. The multi-stage pipeline proposed in FDA likely to be more computationally expensive compared to end-to-end docking-free approaches. A discussion on the computational efficiency and trade-offs of each method would be beneficial.

**Strengths Contributions:**

1. The proposed benchmark focuses on a novel and crucial problem, that is the binding affinity prediction of modified proteins. The introduction of DAVIS-complete fills a significant gap in existing datasets, and the three benchmarks are well-designed to reflect real-world scenarios.

2. The paper thoroughly evaluates both docking-free and docking-based models across diverse settings, providing clear insights into their strengths and limitations.

3. The codes and datasets are openly released.

---

> ### Author Rebuttal · Authors · 2025-07-30
>
> We appreciate the reviewer’s thorough evaluation and thoughtful questions. Below, we provide our detailed responses to each point raised.
>
> **1. To enable docking-free, sequence-based models to handle modified residues, the original vocabulary likely requires adaptation. For example, it should be clarified whether modified amino acids (e.g., phosphotyrosine) are treated as distinct tokens from their unmodified counterparts (e.g., tyrosine). A more detailed description of how modifications are represented in input sequences would strengthen the paper.**
>
> We thank the reviewer for this insightful comment. As described in the **Input Preprocessing section** of the supplementary material, phosphorylation events are currently not explicitly modeled in docking-free methods; modified residues are treated as their unmodified counterparts. We agree that representing phosphorylated residues as distinct tokens is an interesting and valuable suggestion, and we will incorporate this clarification and discuss this potential improvement in the revised version.
>
> **2. The multi-stage pipeline proposed in FDA likely to be more computationally expensive compared to end-to-end docking-free approaches. A discussion on the computational efficiency and trade-offs of each method would be beneficial.**
>
> We thank the reviewer for this important comment regarding computational efficiency. As discussed in the Limitations section of the FDA paper [1], while the Folding-Docking-Affinity (FDA) framework introduces additional computational steps compared to end-to-end docking-free methods, its trade-offs are justified by improved generalizability in challenging scenarios. Specifically, the protein folding step dominates the computational cost, with an average runtime of \~540 seconds per protein whereas docking (~12 seconds) and affinity prediction (\~0.01 seconds) are relatively lightweight. Importantly, in many practical applications, predicted protein structures are readily available from resources such as the AlphaFold database [2], eliminating the need to perform folding for each protein. Finally, as noted in the paper, ongoing advances—such as optimization of stochastic differential equation solvers, reduced diffusion sampling steps, and parallelization strategies—are expected to substantially reduce docking time. Thus, we believe the current trade-off between computational cost and improved generalizability is reasonable, and we will clarify this discussion in the revised version.
>
> We appreciate the reviewer’s thoughtful feedback again and hope our clarifications address the concerns raised.
>
> [1] Wu, M. H., Xie, Z., & Zhi, D. (2025). A Folding-Docking-Affinity framework for protein-ligand binding affinity prediction. Communications Chemistry, 8(1), 1-9.
>
> [2] Varadi, M., Anyango, S., Deshpande, M., Nair, S., Natassia, C., Yordanova, G., ... & Velankar, S. (2022). AlphaFold Protein Structure Database: massively expanding the structural coverage of protein-sequence space with high-accuracy models. Nucleic acids research, 50(D1), D439-D444.

---

> > ### Comment · Reviewer_bWgu · 2025-08-09
> >
> > Thanks for the detailed responses! I have no significant concerns now and will keep my original score for acceptance. Plus, I recommend the authors to clarify the definition of the vocabulary and the trade-offs in docking-based methods in the revised version.

---

> ### Comment · Area_Chair_weB4 · 2025-08-09
>
> Dear Reviewer bWgu,
>
> Please go over the authors' rebuttal and submit your final rating.
>
> Best,
>
> AC

---

### Official Review · Reviewer_9L85 · 2025-07-02

**Rating:** 4
**Confidence:** 3

**Summary:**

The authors propose a modified version of the DAVIS dataset, called DAVIS-complete, that is complementing the existing kinase data with modification-aware versions of protein-ligand pairs, namely 4,032 pairs including substitutions, insertions, deletions, and phosphorylation events.
Leveraging the expanded dataset they introduce three benchmark tasks: Augmented Dataset Prediction, Wild-Type to Modification Generalization, and Few-Shot Modification Generalization. The goal is to asses robustness of models under protein modifications.
Finally they evaluate a set of docking-free and docking-based methods over the three tasks showing different performance based on the model family.

**Dataset Code Accessibility:**

Yes

**Dataset Code Comments:**

Code provided is accessible here: https://github.com/ZhiGroup/DAVIS-complete.
It can be used to run the four experiments for the three benchmarking settings presented in the paper. Besides the old versions of python the code is usable and the instructions are easy to follow.

**Ethical Considerations:**

No, there are no or only very minor ethics concerns

**Final Justification:**

I appreciated the explanations and the additional experiments with stricter splitting strategy. I updated my score accordingly.

**Limitations Weaknesses:**

While interesting for a specific scientific community, the contribution represents an extension of an existing dataset with limited impact.
The limited size and the narrow applicability does not make it extremely relevant or interesting for modern data-intensive machine/deep learning models.
Beyond this impact/relevance considerations, a fundamental shortcoming of the currently proposed dataset is the splitting strategy that is way too lenient risking to alter the purpose of the suggested downstream tasks.
Seq-id is the only method actually splitting based on some similarity notion.
All the others techniques are always risking to put in the same split wild-types and potentially small modifications of themselves. While this can be interesting in some specific tasks, this decision can greatly hinder generalization.
Moreover, to be even stricter it would be ideal to also define splits based on ligand similarity (e.g., using Tanimoto similarity) to make sure test actual generalization to unseen pairs (this is especially critical in the Both-new setting).

**Strengths Contributions:**

The authors tackle an interesting problem that is currently not well represented in existing protein-ligand interaction datasets.
The effort of manually annotating and validating the additional ~4k pairs is remarkable and the benchmarking effort to run a large variety of models is a useful contribution in the domain of chemo-informatics.

---

> ### Author Rebuttal · Authors · 2025-07-30
>
> We appreciate the reviewer’s thorough evaluation and thoughtful questions. Below, we provide our detailed responses to each point raised.
>
> **While interesting for a specific scientific community, the contribution represents an extension of an existing dataset with limited impact. The limited size and the narrow applicability does not make it extremely relevant or interesting for modern data-intensive machine/deep learning models.**
>
> We appreciate the reviewer’s comment. However, we respectfully disagree with the claim that our extension of the DAVIS dataset has limited impact or relevance for modern machine/deep learning models due to its size or applicability. Below, we address these concerns:
> - **Significant Contribution Beyond a Simple Extension**: The curated DAVIS dataset, with 4,032 kinase-ligand pairs incorporating protein modification addresses a critical gap in the field. As stated in the **Introduction**, "current AI-driven models only focus on wild-type proteins, overlooking modified protein versions or applying a simplistic ‘one-size-fits-all’ approach to variants within datasets like DAVIS. This oversight creates a significant gap in understanding how these models perform in real-world biological contexts, where proteins naturally undergo structural or chemical modifications. Models trained solely on such data may overfit to wild-type proteins and fail to generalize to more complex, yet practical, scenarios". By including these modifications, the dataset enables models to better reflect biological realities, advancing precision medicine by addressing real-world protein variants.
>
> - **Dataset Size**: While we understand the concern regarding dataset size, we believe it overlooks the contextual significance and quality of the data. The extended dataset includes 4,032 kinase-ligand binding affinity measurements, which we argue is substantial given the experimental challenges and cost associated with generating such data. Unlike domains such as natural language, image, or video, where large-scale datasets are relatively easy to curate, high-quality biochemical assay data—especially for modified kinases—is scarce and difficult to obtain. Moreover, quality often outweighs quantity in scientific applications. The DAVIS dataset is known for its high experimental homogeneity and high quality Kd measurements, which ensures consistency across data points. This quality enables more trustworthy model evaluation and benchmarking. We have elaborated on this in **Section 2.4**.
>
> - **Relevance to Machine/Deep Learning**: While the dataset may not be large enough to train deep learning models from scratch, it serves as a critical benchmark set for evaluating models trained on other datasets—specifically their ability to capture subtle binding affinity changes induced by protein modifications. Additionally, it supports fine-tuning of pre-trained models, enhancing their ability to generalize to realistic and biologically relevant scenarios.
>
> Lastly, we would like to point out that reviewer nS2t expressed appreciation for this direction, recognizing the importance of expanding a high-quality dataset with biologically relevant kinase modifications. We hope this clarification helps address this concern.
>
> **Beyond this impact/relevance considerations, a fundamental shortcoming of the currently proposed dataset is the splitting strategy that is way too lenient risking to alter the purpose of the suggested downstream tasks. Seq-id is the only method actually splitting based on some similarity notion. All the others techniques are always risking to put in the same split wild-types and potentially small modifications of themselves. While this can be interesting in some specific tasks, this decision can greatly hinder generalization. Moreover, to be even stricter it would be ideal to also define splits based on ligand similarity (e.g., using Tanimoto similarity) to make sure test actual generalization to unseen pairs (this is especially critical in the Both-new setting).**
>
> We thank the reviewer’s feedback regarding the data splitting strategy used in the Augmented Dataset Prediction benchmark. We acknowledge the concern about the lenient nature of the current splitting methods, which could potentially overestimate the generalization performance. We want to clarify that the primary goal of the "Augmented Dataset Prediction" section is specifically to evaluate the influence of dataset enrichment—particularly the inclusion of previously overlooked protein modifications—on predictive performance. This benchmark intentionally follows the exact data splitting strategies utilized by a previous study published in nature machine intelligence [1] to ensure direct and fair comparisons. Indeed, we found that the performance rankings of the models roughly remain consistent with those reported in previous studies [1,2].
>
> Nevertheless, we acknowledge that the current splitting strategies are relatively lenient—particularly with the inclusion of modified proteins—and we fully agree with the reviewer that stricter splits based on protein and ligand similarity would better assess a model’s generalization capability. We are actively addressing this recommendation and have already begun implementing more rigorous and comprehensive splitting strategies as outlined below:
>
> - For drugs, our current strategy treats different drug names as distinct, ensuring no drug name overlap between training and test sets (**new-drug name**). A stricter strategy, as suggested by the reviewer, involves ensuring that drugs in the test set have a Tanimoto similarity ≤ 0.5 (calculated using Morgan fingerprints) with any drug in the training set. We refer to this as the **new-drug structure** split.
>
> - For proteins, we consider three levels of splitting:
>     - **New-protein modification**: Different modifications of the same kinase are treated as distinct unseen proteins (our current approach).
>     - **New-protein name**: If any variant (including wild-type) of a protein appears in training, all its forms are excluded from the test set.
>     - **New-seq-id protein**: Based on sequence identity thresholds (≤ 0.5), as already implemented in our paper, and considered the strictest.
>
> Combining these strategies results in six configurations for the "both-new" split as the following table. The current configuration presented in the paper is (**new-protein modification & new-drug name**), while the strictest setting, as suggested by the reviewer, is (**new-seq-id protein & new-drug structure**).
>
> |                           | **new-drug name** (current)    | **new-drug structure** (update)      |
> |-------------------|-----------------------|---------------------------|
> | **new-protein modification (current)** |      (current)    |  -                                    |
> | **new-protein name (update)**          |      -                |    -                                  |
> | **new-seq-id protein (current)**           |    -                   |   (update)                   |
>
> Due to time constraints, we have conducted preliminary experiments under stricter settings using two models: DeepDTA and MGraphDTA. These evaluations were performed on the **new-drug structure**, **new-protein name**, and the combined (**new-seq-id protein & new-drug structure**) settings. The result as the following
>
> | Model        |            new-drug structure|            |              new-protein name|              |   new-seq-id protein & new-drug structure|   |
> |--------------|----------------------|----------------------|----------------------|----------------------|----------------------|----------------------|
> |              | MSE                 | Rp                  | MSE                 | Rp                  | MSE                 | Rp                  |
> | DeepDTA      | 0.69 (0.06)         | 0.26 (0.06)         | 0.42 (0.04)         | 0.70 (0.03)         | 0.86 (0.16)         | 0.17 (0.15)         |
> | MGraphDTA    |  0.76  (0.07)       | 0.24 (0.08)                     |  0.38 (0.03)                   |    0.72 (0.05)               |  1.01 (0.28)                   |   0.22 (0.12)            |
>
> As expected, these stricter splitting strategies resulted in a noticeable performance drop compared to the initial configuration shown in **Table 2**. We are in the process of completing the full evaluation for all models across all six configurations and will include the expanded results and analysis in the revised version of the paper.
>
> However, we argue that the initial splitting approach should be considered methodologically imprecise at most and addressable rather than a fundamental shortcoming. We believe the splitting decision does not jeopardize our overall contribution; furthermore, even with stricter splitting evaluations, our primary contribution—introducing a complete, modification-aware dataset and corresponding benchmarks—would remain unchanged and valid.
>
> We appreciate the reviewer’s thoughtful feedback again and hope our clarifications address the concerns raised.
>
> [1] Luo, Y., Liu, Y., & Peng, J. (2023). Calibrated geometric deep learning improves kinase–drug binding predictions. Nature machine intelligence, 5(12), 1390-1401.
>
> [2] Wu, M. H., Xie, Z., & Zhi, D. (2025). A Folding-Docking-Affinity framework for protein-ligand binding affinity prediction. Communications Chemistry, 8(1), 1-9.

---

> > ### Comment · Reviewer_9L85 · 2025-08-05
> >
> > Thanks, I appreciated the explanations and the additional experiments with stricter splitting strategy. I updated my score accordingly.

---

### Official Review · Reviewer_nS2t · 2025-07-07

**Rating:** 5
**Confidence:** 3

**Summary:**

The paper addresses a significant limitation in protein-ligand binding affinity prediction by recognizing that prevailing AI models tend to overfit to simplified datasets that lack biologically relevant protein modifications. To address this, the authors introduce DAVIS-complete, a comprehensive and modification-aware extension of the widely used DAVIS dataset, which now includes 4,032 additional kinase–ligand pairs featuring substitutions, insertions, deletions, and phosphorylation events, expanding the dataset to 31,824 entries. By systematically evaluating five docking-free models and one docking-based model (FDA) on these enriched benchmarks, the study exposes the shortcomings of current approaches, particularly their tendency to overfit to wild-type proteins. It demonstrates that strategies such as fine-tuning can enhance model performance. These advancements are anticipated to lay the groundwork for developing predictive models that more effectively generalize to protein modifications, thereby supporting progress in precision medicine and drug discovery.

**Dataset Code Accessibility:**

Yes

**Dataset Code Comments:**

The generated dataset is available on **Zenodo**.
**GitHub** has clear instructions on how to download the data and reproduce the results shown in the paper, along with all the necessary scripts.

**Ethical Considerations:**

No, there are no or only very minor ethics concerns

**Final Justification:**

Since the authors took care of all the weaknesses mentioned to their best effort, I updated my rating to 5.

**Limitations Weaknesses:**

**1. Lack of Analysis on the Impact of Modifications on Binding Affinity**
The authors did not systematically evaluate how different modifications to the proteins influence binding affinity. This omission may confound the assessment of the model's generalization capability. Specifically, if certain modifications have only minimal effects on affinity, the model's apparent ability to generalize could be attributed to its prior exposure to the original protein sequences, rather than to true generalization to novel modifications.

**2. Ambiguity in Data Splitting Regarding Unseen Proteins**
While the authors state that kinase modifications are treated as unseen proteins in the data split, it is unclear whether the dataset also includes genuinely unseen proteins that are not present in any form during training. Clarification is needed regarding the presence of such proteins. Furthermore, it is important to evaluate and report model performance separately for completely unseen proteins versus proteins with unseen modifications, to accurately assess the model's ability to generalize to novel protein sequences.

**3. Dataset Balance and Affinity Distribution**
The manuscript does not address whether the dataset is balanced with respect to high and low affinity pairs. An imbalance, particularly an overrepresentation of lower affinity (higher Kd value) pairs, could bias the model towards learning the average affinity and performing better on these cases. The authors should analyze the distribution of affinity values and report whether predictive performance differs between high- and low-affinity pairs, to ensure that the model is not simply optimizing for the most common affinity range in the dataset.

**Strengths Contributions:**

**High-Quality, Experimentally Consistent Dataset**
A key strength of this work is its foundation on the DAVIS dataset, which is recognized for its high experimental consistency and quality in measuring kinase–ligand interactions through dissociation constants (Kd). By leveraging DAVIS, the authors minimize variability arising from heterogeneous experimental conditions, a common challenge in datasets such as BindingDB. The meticulous manual curation of DAVIS-complete, including the mapping of Entrez Gene Symbols to UniProt IDs and thorough sequence retrieval and annotation, further underscores the authors’ commitment to data integrity and reliability.

**Introduction of Novel and Biologically Relevant Benchmarks**
The paper introduces three thoughtfully designed benchmark settings: Augmented Dataset Prediction, Wild-Type to Modification Generalization, and Few-Shot Modification Generalization. Each benchmark is closely aligned with practical drug discovery scenarios, enabling the assessment of general predictive performance, the ability to generalize to unseen protein modifications (zero-shot), and the adaptability of models with limited modified data—an important consideration for precision medicine. The inclusion of rigorous baselines, such as "wild-type ground truth" (WT-GT) and "wild-type prediction" (WT-Pred), in the Wild-Type to Modification Generalization benchmark further strengthens the evaluation by providing clear standards for assessing model sensitivity to modifications.

**Strong Presentation and Commitment to Reproducibility**
The manuscript is clearly structured, guiding the reader logically from the problem statement through the methodology, results, and discussion of limitations. Visual elements, such as Figure 1, effectively clarify complex processes, including dataset curation and benchmark construction, while tables (e.g., Tables 2–5) present results with comprehensive statistical reporting. Importantly, the authors ensure reproducibility by making all data and code openly available via GitHub, thus promoting transparency and facilitating further research within the community.

---

> ### Author Rebuttal · Authors · 2025-07-30
>
> We appreciate the reviewer’s thorough evaluation and thoughtful questions. Below, we provide our detailed responses to each point raised.
>
> **1. Lack of Analysis on the Impact of Modifications on Binding Affinity**:
>
> **The authors did not systematically evaluate how different modifications to the proteins influence binding affinity. This omission may confound the assessment of the model's generalization capability. Specifically, if certain modifications have only minimal effects on affinity, the model's apparent ability to generalize could be attributed to its prior exposure to the original protein sequences, rather than to true generalization to novel modifications.**
>
> We appreciate the reviewer highlighting this critical concern and fully acknowledge the importance of systematically evaluating how different protein modifications impact binding affinity. To address this point, we have now conducted a quantitative analysis showing that the magnitude of affinity changes (ΔpKd) ranges approximately from –4.49 to +3.02. However, as noted in **Section 6**, a key limitation of the DAVIS dataset is that binding affinity measurements (Kd) are capped at values greater than 10 µM. As a result, for roughly half of the modified protein-ligand pairs—where both the wild-type and the modified proteins exhibit affinities beyond this threshold—we lose track of the exact magnitude of modification-induced changes. For the trackable alternations (ΔpKd ≠ 0), we summarize the distribution of pKd changes as the following:
> |        | Mean  | Std   | Min   | 25%   | 50%   | 75%   | Max   |
> |--------|-------|-------|-------|-------|-------|-------|-------|
> | Value  | -0.21 | 0.84  | -4.49 | -0.59 | -0.04 | 0.30  | 3.02  |
>
> We agree with the reviewer that including untrackable pairs may lead to an overestimation of generalization performance, as models could simply memorize the wild-type values without capturing the true effects of the modifications. To better understand this impact, we computed performance metrics separately for the trackable and untrackable subsets in the **Global Modification Generalization** benchmark. The results are shown in the table below.
> | Model        | Trackable                        |                          |                          | Untrackable                      |                          |                          |
> |--------------|----------------------------------|--------------------------|--------------------------|----------------------------------|--------------------------|--------------------------|
> |              | MSE                              | Rp                       | C-index                  | MSE                              | Rp                       | C-index                  |
> | DeepDTA      | 0.71 (0.00)                      | 0.80 (0.01)              | 0.79 (0.00)              | 0.01 (0.01)                      | -                        | -                        |
> | AttentionDTA | 0.72 (0.04)                      | 0.80 (0.01)              | 0.79 (0.00)              | 0.01 (0.00)                      | -                        | -                        |
> | GraphDTA     | 1.20 (0.05)                      | 0.66 (0.01)              | 0.73 (0.01)              | 0.05 (0.02)                      | -                        | -                        |
> | DGaphDTA     | 0.75 (0.04)                      | 0.80 (0.01)              | 0.79 (0.00)              | 0.01 (0.00)                      | -                        | -                        |
> | MGraphDTA    | 0.71 (0.02)                      | 0.80 (0.00)              | 0.79 (0.00)              | 0.01 (0.01)                      | -                        | -                        |
> | FDA          | 1.26 (0.01)                      | 0.65 (0.01)              | 0.73 (0.00)              | 0.05 (0.00)                      | -                        | -                        |
>
> As expected, the majority of prediction error originates from the trackable subset, while the untrackable subset is predicted with near-perfect accuracy. The near-perfect prediction on untrackable data dilutes the error from the trackable subset, leading to an inflated overall performance, shown in the **Table 3**. We will include this analysis and its implications in the revised version to help provide a clearer understanding of the benchmark results and ensure a more accurate interpretation of the model’s generalization performance.
>
> On the other hand, we focused our analysis in the “Same-ligand, different-modifications” and “Same-modification, different-ligands” subtasks (**Sections 5.2 and 5.3**) only on the trackable subset, in order to better capture modification-induced changes beyond potential memorization effects.
>
> **2. Ambiguity in Data Splitting Regarding Unseen Proteins**
>
> **While the authors state that kinase modifications are treated as unseen proteins in the data split, it is unclear whether the dataset also includes genuinely unseen proteins that are not present in any form during training. Clarification is needed regarding the presence of such proteins. Furthermore, it is important to evaluate and report model performance separately for completely unseen proteins versus proteins with unseen modifications, to accurately assess the model's ability to generalize to novel protein sequences.**
>
> We thank the reviewer for highlighting this important point. We acknowledge that our description of the unseen protein split was ambiguous, and that the current setting—treating different modifications of the same kinase as unseen proteins—is not sufficiently rigorous and may lead to an overestimation of generalization performance. For example, under the current setting, the model might be trained on ABL1(Q252H) and tested on ABL1(T315I). If both modifications have minimal impact on binding affinity, the model could memorize the binding value, resulting in inflated performance. We fully agree with the reviewer’s suggestion and have accordingly defined three levels of protein-based splitting:
>
> - **New-protein modification**: Different modifications of the same kinase are treated as distinct unseen proteins (our current approach).
> - **New-protein name**: If any variant (including wild-type) of a protein appears in training, all its forms are excluded from the test set.
> - **New-seq-id protein**: Based on sequence identity thresholds (≤ 0.5), as already implemented in our paper, and considered the strictest.
>
> To begin addressing this issue, we conducted preliminary experiments using two models—GraphDTA and MGraphDTA—under the stricter new-protein name settings. Due to time constraints, we have not yet evaluated all models, but we will include the complete results in the revised version. The preliminary results are shown in the following table:
>
> | Model      | new-protein modification (current) |                              | new-protein name (update)     |                              | new-seq-id protein (current) |                              |
> |------------|-------------------------------------|------------------------------|-------------------------------|------------------------------|-------------------------------|------------------------------|
> |            | MSE                                 | Rp                           | MSE                           | Rp                           | MSE                           | Rp                           |
> | DeepDTA    | 0.31 (0.05)                         | 0.80 (0.03)                  | 0.42 (0.04)                   | 0.70 (0.03)                  | 0.64 (0.08)                   | 0.62 (0.05)                  |
> | MGraphDTA  | 0.30 (0.02)                         | 0.80 (0.02)                  | 0.38 (0.03)                   | 0.72 (0.05)                  | 0.62 (0.10)                   | 0.62 (0.03)                  |
>
>
> **3. Dataset Balance and Affinity Distribution**
>
> **The manuscript does not address whether the dataset is balanced with respect to high and low affinity pairs. An imbalance, particularly an overrepresentation of lower affinity (higher Kd value) pairs, could bias the model towards learning the average affinity and performing better on these cases. The authors should analyze the distribution of affinity values and report whether predictive performance differs between high- and low-affinity pairs, to ensure that the model is not simply optimizing for the most common affinity range in the dataset.**
>
> We thank the reviewer for raising this important point. Indeed, the DAVIS dataset, despite its experimental homogeneity, is well-known for its highly imbalance in binding affinity distributions. Specifically, around 70% of the protein–ligand pairs have Kd values capped at 10 µM (pKd=5), resulting in an overrepresentation of lower-affinity interactions. The distribution of non-capped affinity values is summarized in the following table:
>
> |  | Mean | Std  | Min  | 25%  | 50%  | 75%  | Max   |
> |-----------|------|------|------|------|------|------|--------|
> | Value     | 6.48 | 1.05 | 5.00 | 5.68 | 6.24 | 7.08 | 10.80 |
>
> We have already described this challenge in the **Section 6**. Meanwhile, we agree with the reviewer’s suggestion that a deeper analysis of predictive performance differences between high- and low-affinity pairs is necessary. To comprehensively address this, we plan to perform an explicit subgroup evaluation based on affinity ranges, clearly reporting model performance separately for high-affinity and low-affinity interactions in the revised version.
>
> We appreciate the reviewer’s thoughtful feedback again and hope our clarifications address the concerns raised.

---

> > ### Comment · Reviewer_nS2t · 2025-08-02
> > **Response to Rebuttal**
> >
> > Thanks for handling all the comments mentioned in my review. I updated the rating accordingly.

---

### Decision · Program_Chairs · 2025-09-18

**Decision:**

Accept (poster)

**Comment:**

**Summary**:

This paper addresses a critical limitation in protein–ligand binding affinity prediction, namely that existing AI models often overfit to simplified datasets that lack biologically relevant protein modifications. To overcome this, the authors introduce DAVIS-complete, a modification-aware extension of the widely used DAVIS dataset. The new resource incorporates 4,032 additional kinase–ligand pairs with substitutions, insertions, deletions, and phosphorylation events, expanding the dataset to 31,824 entries. Using this benchmark, the authors systematically evaluate five docking-free models and one docking-based model (FDA), revealing that current approaches tend to overfit to wild-type proteins. They further show that strategies such as fine-tuning can improve generalization, underscoring the importance of modification-aware benchmarks for advancing predictive models in precision medicine and drug discovery.

**Strengths**:

- The dataset is a straightforward extension of an established benchmark, which facilitates adoption by the community.

- The authors carefully design train–test splits to reflect realistic challenges, such as predicting the effects of protein modifications.

- The dataset is described with mathematical rigor (Section 3), improving transparency and clarity.

- The authors benchmark multiple models on these new splits, providing an informative and comparative evaluation.

**Weaknesses**:

- As raised during the rebuttal, the new results should be more thoroughly integrated into the paper to provide clearer insights into benchmark performance and model generalization.

- The revised version should clarify the vocabulary definition and better discuss the trade-offs of docking-based methods, as suggested by the reviewers.

During the discussion, reviewers agreed that this is a strong and impactful contribution. Despite minor revisions required for clarity and completeness, the consensus is that the paper’s strengths outweigh its weaknesses. I therefore recommend acceptance.